# Cytochrome P450 Metabolism of Polyunsaturated Fatty Acids and Neurodegeneration

**DOI:** 10.3390/nu12113523

**Published:** 2020-11-16

**Authors:** Morteza Sarparast, Devon Dattmore, Jamie Alan, Kin Sing Stephen Lee

**Affiliations:** 1Department of Chemistry, Michigan State University, East Lansing, MI 48824, USA; sarparas@msu.edu; 2Department of Pharmacology and Toxicology, Michigan State University, East Lansing, MI 48824, USA; dattmore@msu.edu

**Keywords:** polyunsaturated fatty acid, cytochrome P450, epoxy-polyunsaturated fatty acid, neurodegeneration, neurodegenerative disease, dietary intervention

## Abstract

Due to the aging population in the world, neurodegenerative diseases have become a serious public health issue that greatly impacts patients’ quality of life and adds a huge economic burden. Even after decades of research, there is no effective curative treatment for neurodegenerative diseases. Polyunsaturated fatty acids (PUFAs) have become an emerging dietary medical intervention for health maintenance and treatment of diseases, including neurodegenerative diseases. Recent research demonstrated that the oxidized metabolites, particularly the cytochrome P450 (CYP) metabolites, of PUFAs are beneficial to several neurodegenerative diseases, including Alzheimer’s disease and Parkinson’s disease; however, their mechanism(s) remains unclear. The endogenous levels of CYP metabolites are greatly affected by our diet, endogenous synthesis, and the downstream metabolism. While the activity of omega-3 (ω-3) CYP PUFA metabolites and omega-6 (ω-6) CYP PUFA metabolites largely overlap, the ω-3 CYP PUFA metabolites are more active in general. In this review, we will briefly summarize recent findings regarding the biosynthesis and metabolism of CYP PUFA metabolites. We will also discuss the potential mechanism(s) of CYP PUFA metabolites in neurodegeneration, which will ultimately improve our understanding of how PUFAs affect neurodegeneration and may identify potential drug targets for neurodegenerative diseases.

## 1. Introduction

Neurodegenerative diseases (NDs) are affected by both genetic and environmental factors suggesting that there are likely multiple etiologies for these diseases [1,2]. In addition, the prevalence of NDs correlates well with age [3]. According to the United Nations, the population over the age of 65 is expected to increase from approximately 9% (2019) to roughly 20% by 2050. With this demographic change, a coinciding increased incidence of age-related NDs is expected in the near future [4]. Despite decades of effort, no curative treatment has been developed for these diseases, and almost all medication interventions are aimed at reducing the symptoms. Perhaps the primary reason for the lack of treatments for NDs is that the underlying mechanism(s) for ND pathologies has yet to be identified. Human genetic studies revealed several genes responsible for NDs, such as apolipoprotein E (APOE), which has been extensively reviewed and is not the focus of this review [4,5]. On the other hand, numerous studies reveal that environmental factors or a complex interaction between environmental and genetic factors result in slow and sustained dysfunctions in the nervous system during aging and could be major causes of NDs. Among the environmental factors, exposure to pesticides and trace metals, head injuries, lifestyle, and diet, are deemed to be most influential to ND risk [6]. Interestingly, one pathway that appears to be involved in the aging process, but remains to be fully explored, is the metabolism of polyunsaturated fatty acids (PUFAs). In this review paper, we focus primarily on dietary factors, specifically, the relationship between cytochrome P450 (CYP) metabolism of polyunsaturated fatty acids (PUFAs) and age-associated NDs [1,2].

In mammals, ω-3 PUFAs cannot be synthesized endogenously; therefore, they must be obtained from dietary sources [7]. PUFAs are suspected to play significant roles in neural functions due to their abundance in neural tissues. Arachidonic acid (AA) and docosahexaenoic acid (DHA) are the two most abundant PUFAs within the nervous system and together comprise roughly 35% of the lipid content in the brain tissue [8]. It has been demonstrated that dietary PUFA intake is beneficial for neurodevelopment and attenuating neurodegeneration [9]. The Rotterdam study, which assessed a cohort of 5289 subjects aged 55 and older, showed an association between PUFA intake and reduced incidence of PD [10]. In rodents fed with a diet deficient in ω-3 PUFAs, a dramatic decrease in brain PUFA content have been observed, which were also accompanied with a decrease in the numbers of dopaminergic neurons in the substantia nigra [11]. Manipulating the PUFA composition of the cell membrane has been shown to change the function and/or signaling of a variety of receptors, including cholinergic, dopaminergic, and GABAergic receptors [12], while the detailed mechanism remains largely unknown. The endogenous level of PUFAs greatly affected the composition of their downstream metabolites in vivo. Therefore, it has been suggested that the downstream metabolites may be partly responsible for the action of ω-3 and ω-6 PUFAs in neurons.

PUFAs are metabolized by three main oxidative pathways, which include (i) lipoxygenase (LOX), (ii) cyclooxygenase (COX), and (iii) cytochrome P450 (CYP) pathways to produce different oxidized lipid mediators called oxylipins [13]. Generally, ω-6 PUFA oxylipins tend to be proinflammatory, while ω-3 PUFA oxylipins are considered anti-inflammatory and pro-resolving. Oxylipins are mediators in physiological processes in various tissues including neurons [13]. AA oxylipins, such as lipoxin A4, as well as prostaglandin D2 and E2, have shown neuroprotective properties [13]. The pro-resolving mediators derived from DHA such as protectin D1, as well as resolvin D1 and D2, seem to inhibit age-related memory decline and protect the brain from cell injury and death [13,14]. Furthermore, the oxylipins generated by COX and LOX, such as prostaglandin and leukotrienes, respectively, tend to be pro-inflammatory and exert excitatory effects on neurons. On the other hand, epoxy-PUFAs (Ep-PUFAs), oxylipins generated by CYP enzymes, seem to produce the opposite effects, and are neuroprotective, anti-hypertensive, and analgesic [13,15,16]. While the activity of ω-3 Ep-PUFAs and ω-6 Ep-PUFAs largely overlap, the ω-3 Ep-PUFAs are generally more active than ω-6 Ep-PUFAs. The beneficial properties of Ep-PUFAs appear to be diminished when Ep-PUFAs are converted to their corresponding 1,2-diols by soluble epoxy hydrolase (sEH) [17], which will be discussed later in this review. As more research has accumulated, CYP metabolites of PUFAs have been suggested to be a key class of oxylipins that affect neurodegeneration. Inhibition of sEH, the enzyme largely responsible for the degradation of CYP PUFA metabolites, has been shown to be beneficial in Alzheimer’s disease (AD) and Parkinson’s disease (PD) [18]. Additionally, inhibition or the genetic knockout of sEH can protect the dopaminergic neurons in the mouse brain against neurotoxins [19]. Even though there is a controversy, in randomized clinical trials studying the effects of ω-3 PUFAs in AD, most observational studies have shown the beneficial effects of ω-3 intake on reducing the incidence of AD [9,13,18]. Additionally, the oxylipin profiles of blood serum in AD subjects show around 20% higher levels of dihydroxyeicosatrienoic acid (which is the product of the sEH metabolism of the epoxy-metabolite of AA) as compared to the elderly individuals that were cognitively healthy [20]. Therefore, sEH is an important enzyme in PUFA metabolism and a lucrative drug target for NDs [21]. However, the mechanism of action of the CYP PUFA metabolites remains largely unknown. The endogenous level of CYP PUFA metabolites is not only affected by their metabolism but is also greatly affected by the endogenous level of their PUFA precursors as well as their uptake by diet [22,23]. Therefore, in this review, we will: (1) describe the biosynthesis and metabolism of PUFAs and downstream CYP metabolites and the role of biosynthesis and metabolism of PUFAs in neurodegeneration; (2) highlight the key CYP and EH enzymes present in central nervous system (CNS); and (3) elaborate on the potential mechanism(s) of action of CYP PUFA metabolites in neuroinflammation and corresponding neurodegeneration.

## 2. Neurodegeneration: A Brief Overview

Neurons have an incredibly high respiratory rate due to the presence of highly active processes such as vesicle trafficking, neurotransmitter synthesis, protein synthesis, molecular/ion transport, etc., and thus generate a great deal of oxidative stress [1,24]. This further necessitates a high demand of energy for the maintenance of redox homeostasis and, organelle and protein quality control [1,24]. Chronic and excessive perturbations in these pathways can cause neurodegeneration. Figure 1 displays common neuronal pathways that are disturbed in neurodegenerative diseases [1,25]. These pathways are mostly interdependent. For instance, chronic neuroinflammation can lead to protein accumulation, endoplasmic reticulum (ER) stress, mitochondria dysfunction, uncontrolled oxidative stress, axonal transport impairment, and apoptosis, which are detrimental to neurons, leading to NDs [26,27]. The major ND pathologies include, but are not limited to protein misfolding, synaptic dystrophy, changes in neurotransmitter production, an increased oxidative stress response, and loss of neurons [2,28,29,30]. In addition, reduction in brain volume is often observed (in both gray and white matter), along with increased lesions in white matter and dysfunction of the blood–brain barrier (BBB) [28]. While the mechanisms that are involved in the pathology of NDs are pertinent to this review, the details are not included here, as they have been extensively discussed in other reviews [1,24,25]. Our focus here is to review and discuss how CYP metabolism of PUFAs is potentially involved in neurodegeneration.

## 3. Overview of PUFAs

PUFAs are long-chain fatty acids comprising of at least two carbon-carbon double bonds that play crucial roles in a variety of physiological processes. The endogenous levels of ω-3 and ω-6 PUFAs significantly modulate the in vivo levels of their downstream metabolites, particularly the CYP PUFA metabolites [31,32]. In mammals, small quantities of PUFAs can be synthesized endogenously, and the rest are obtained from the diet. An overview of the the biological levels of PUFAs, and thus their downstream metabolism is presented in this section.

### 3.1. Biosynthesis of PUFAs and Neurodegenerative Diseases

The major steps in PUFA biosynthesis in organisms are: (i) elongation, by elongase enzymes (elongase of very long chain fatty acids, *ELOVL*); and (ii) desaturation by desaturase enzymes (fatty acid desaturase, *FADS*) [7,33]. Some metazoans, such as *Caenorhabditis elegans*, are capable of endogenous production of ω-6 and ω-3 PUFAs through the conversion of ω-9 monounsaturated oleic acid (OA) to ω-6 linoleic acid (LA) via a Δ12 desaturase, and then further conversion of ω-6 PUFAs to ω-3 PUFAs via Δ15 (ω-3) desaturase [34]. These animals do not require dietary sources of PUFAs. Vertebrates, however, do not have genes encoding functional Δ12 and ω-3 desaturase enzymes, and thus cannot endogenously synthesize PUFAs from OA [35]. Despite this, vertebrates do have the ability to use LA and α-linoleic acid (ALA), which must be obtained from the diet, as precursors for the biosynthesis of the other ω-6 and ω-3 PUFAs, respectively, through desaturation and elongation [7,33]. The conventional enzymatic pathway that produces these PUFAs is depicted in Figure 2A.

Although humans only synthesize a small amount of PUFAs, recent studies provide evidence that FADS in humans play an important role in modulating endogenous levels of different PUFAs. A genome-wide genotyping study conducted by Ameur et al. revealed two common haplotypes of the *FADS* gene: (i) haplotype D that can drive more active conversion in PUFAs, and (ii) the less active haplotype A [36]. People homozygous for the D type have higher levels of AA (43%) and DHA (24%), as well as greater plasma lipid levels compared to those who are homozygous for haplotype A [36]. Several studies have shown the association between breastfeeding and child brain development, as a *FADS* gene variant can control the PUFA composition of breast milk [37,38,39,40]. Furthermore, from rare human studies, a very low level of Δ6 desaturase was found in patients with Sjögren–Larsson syndrome, which is characterized by neurological, skin, and eye problems [41]. A single-nucleotide polymorphism (SNP) in *FADS2* was also discovered to be associated with the occurrence of attention-deficit/hyperactivity disorder (ADHD), which is suggested to be a result from abnormal regulation of DA at the neural synapse [42]. Most of these rare studies can be considered as preliminary data, and replication in a larger population, and consideration of other PUFA metabolic genes such as *ELOVL* are necessary. In addition to human studies, mouse models with genetic defects in *Fads1* and *Fads2* have been conducted [43,44]. The *Fads1* knockout mice failed to thrive and died before reaching 12 weeks of age. These mice also exhibited perturbed immune cell homeostasis and severe inflammatory problems. Dietary supplementation with AA prolonged the lifespan of these mice to levels comparable to wild-type mice [43]. Surprisingly, knockout of the *Fads2* gene in mice did not result in significant impairment in lifespan and viability, but rather produced sterility (in both sexes) and increased bleeding time [44]. Brown and colleagues observed that a selective knockdown of *Fads1* in adult hyperlipidemic mice results in striking changes of both ω-3 and ω-6 PUFA levels and their corresponding proinflammatory and proresolving lipid mediators, suggesting an important regulatory effect of Δ-5 desaturase in inflammation initiation and resolution [45]. Thus, variability in gene expression may be one of the key factors impacting PUFA downstream metabolism by controlling the *Fads/FADS* (in mouse/human) genes. Considering this idea, researchers studying the effects of PUFA supplementation should consider genetic effects in the specific population under study, which might be the reason for the inconsistency found in the clinical and epidemiology studies regarding the PUFA effects on human health and disease.

### 3.2. Dietary PUFA

Although humans make both desaturase and elongase enzymes, the conversion/biosynthesis of PUFAs is quite limited in humans. Because of this, PUFA levels in blood and tissues are modulated by dietary intake [7,46]. Hence, there is a great interest surrounding the potential benefits of supplementation, and ω-3 PUFAs are one of the most consumed dietary supplements [7]. The main sources of fatty acids are different among countries, and mainly controlled by food availability, economy, and culture [7,39]. ω-3 PUFAs primarily originate from plant, algal, marine, and protozoan sources. While plants such as nuts, some seeds, and vegetable oil are the primary source of ALA (Figure 2B), marine animals are the main source of eicosapentaenoic acid (EPA), DHA, and docosapentaenoic acid (ω-3) (DPA3) (Figure 2C) [47]. Dietary supplementation studies in randomized clinical trials show an inconclusive results regarding the effects of PUFAs on neurodegeneration, that could be due to the genetic variability, diet and different lifestyles and habits in the population under study [46]. Furthermore, studies in humans demonstrate a causal relationship between the dietary lipid ratio supplementation of ω-3 and ω-6 PUFAs and NDs, such as AD and PD, which is beyond the scope of this review [6,10,46,48,49].

**Figure 2 nutrients-12-03523-f002:**
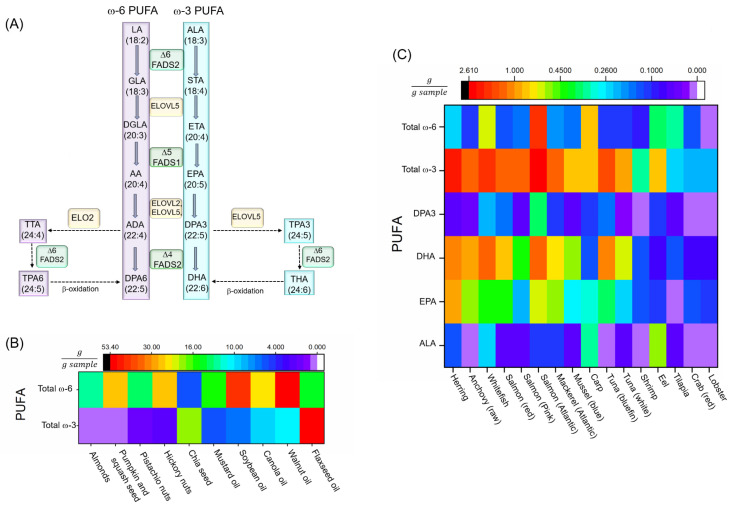
(**A**) PUFA biosynthesis pathways. (**B**) Common vegetable oil or seeds. The total ω-PUFA content is almost exclusively ALA, with little to no EPA, DHA, and DPA3 [47]. (**C**) PUFA amounts in different fish and seafood (all are cooked or baked unless otherwise mentioned) [47]. Abbreviations- AA: arachidonic acid, AD: adrenic acid, ALA: α-linolenic Acid, Δ6: delta-4 desaturase, Δ5: delta-5 desaturase, Δ6: delta-6 desaturase, DGLA: dihomo-γ-linolenic acid, DPA3: docosapentaenoic acid (ω-3), DPA6: docosapentaenoic acid (ω-6), DHA: docosahexaenoic acid, ELOVL: elongase of very long chain fatty acids, EPA: eicosapentaenoic acid, ETA: eicosatrienoic acid, FADS 1: fatty acid desaturase 1, FADS2: fatty acid desaturase 2, GLA: γ-linolenic acid, LA: Linoleic acid, PUFA: polyunsaturated fatty acid, THA: tetracosahexaenoic acid, TPA3: tetracosapentenoic acid (ω-3), TPA6: tetracosapentenoic acid (ω-6), TTA: tetracosatetraenoic acid, STA: stearidonic acid.

## 4. CYP: A Key Monooxygenase Enzyme in PUFAs Metabolism

PUFAs are mainly metabolized through three oxidative (pathways: (i) lipoxygenases (LOX), (ii) cyclooxygenases (COX), and (iii) CYPs) to produce lipid signaling molecules called oxylipins [50,51,52]. Most of these mono-oxygenated lipid metabolites are key lipid mediators in physiological processes in mammals. Research regarding the effects of the lipoxygenase and cyclooxygenase pathways on neurodegeneration has been extensively reviewed [52,53]; thus, these are not the focus of this review. In this section, we will summarize the recent findings on a CYP pathway of PUFAs and the effects of CYP selectivity on mammalian physiology. The CYP metabolic pathway of AA is depicted in Figure 3A.

### 4.1. Characteristics of CYP

CYPs are heme-thiolate proteins primarily involved in synthesis and metabolism of xenobiotics, as well as endogenous biological molecules such as steroid hormones, fatty acids, cholesterol, drugs, vitamin D, etc., via oxidation [54]. CYPs, in general, contain a signature residue sequence of FXXGXbXXCXG, in which Xb is a basic residue and the cysteine is located at the axial position to the heme, and a Soret peak at 450 nm when a carbon monoxide binds to the Fe(II) of the heme group [55,56]. Note that there are some other proteins with the same heme group, axial cysteine residue, and similar Soret peak in the presence of carbon monoxide, as well as some related catalytic properties such as some peroxidases and nitric oxide synthases, but they are not considered CYP enzymes. The 3D structures of these proteins also differentiate them from CYP enzymes, which share the same folding [57,58].

The human genome project has identified 57 genes expressing different CYP enzymes, which are grouped into 18 families (43 sub-families) based on the amino acid sequence homology [59,60]. Thus, each CYP enzyme is named by a number representing the family and a letter indicating the subfamily, followed by a second number specific for an individual CYP enzyme (e.g., CYP2J2). In contrast to prokaryotes that have soluble CYP enzymes, in mammals CYPs are primarily membrane-associated proteins located either on ER or mitochondria membranes [61,62]. The catalytic domain of these enzymes is partially immersed in the membrane and can move along the membrane surface. The active site connection to both the cytosolic environment and the membrane through networks of access channels allows them to interact with substrates in either compartment [62]. Of the 57 human CYP enzymes, 50 are located on the ER and are usually involved in xenobiotic metabolism (i.e., drugs and environmental pollutants), while the rest are located in the mitochondria membrane and are generally engaged in the metabolism/biosynthesis of endogenous molecules [63]. Even though these enzymes are mostly expressed in the liver, they can also be expressed in many other tissues including, but not limited to, kidney, brain, intestinal mucosa, skin, and lung [64].

### 4.2. Catalytic Function and Mechanism of CYP

Historically, the first CYP enzyme was described by Klingenberg and Garfinkel as an unknown pigment that binds carbon monoxide in its reduced form and produces a Soret absorption peak at 450 nm [65,66]. This unknown pigment was then identified as a new cytochrome by Omura et al. [55]. About ten years later, in 1979, Benhamou and colleagues further confirmed this observation by studying the inhibitory effect of AA administration on metabolizing the hepatic drug by CYP enzymes in mice [67].

CYPs catalyze a large variety of reactions including oxidation of heteroatoms, heteroatom dealkylation, carbon-carbon bond cleavage, desaturation, ring formation, aryl ring couplings, and rearrangements of oxygenated molecules [68,69,70]. The monooxygenase activity of CYPs has been discussed thoroughly elsewhere [71,72,73], and, therefore, we will only provide a brief description here. Figure 3B illustrates the epoxidation mechanism of CYPs, related to its epoxygenase activity, in seven steps: (1) before binding of the substrate to the CYP protein, there is an equilibrium between the hexa- and pentacoordinate Fe(III); substrate binding to the CYP enzymes shifts the equilibrium in favor of pentacoordinate; (2) an electron will transfer to this complex either directly from NADPH or through a redox protein partner, to reduce Fe (III) to Fe (II) (note that this step is critical for substrate oxidation, as the diatom oxygen cannot bind to Fe (III)); (3) oxygen binds to the Fe(II); (4) the next electron is either transferred directly from NAD(P)H or through a redox protein partner; (5) two subsequent protonations occur; (6) the complex gets deprotonated by releasing a water molecule, which results in an iron-oxo complex; (7) finally, the oxygen atom is transferred to the substrate and results in an oxidized product [71,72,73].

### 4.3. Major CYP Responsible for PUFA Metabolism

CYPs metabolize PUFAs to either Ep-PUFAs (epoxygenase activity) or hydroxy-PUFAs (with hydroxylase activity). The final product depends on the specific CYP enzyme, as well as the type of PUFA substrate. In general, the regio/stereoselectivity, catalytic properties, and structure of each CYP enzyme might be significantly determined by a B-C loop in their structure, controlling the type of reaction and prevalence of a specific regio/stereoisomer product [56,74,75]. In this section, we discuss the major CYPs and their products when ω-3 and ω-6 PUFAs are substrates.

#### 4.3.1. Regio- and Stereoselective Epoxidation by Major CYPs

In the case of AA, the main CYPs that generate Ep-PUFAs are CYP2B, 2C, and 2J sub-families (known as AA epoxygenase), while 1A, 4A, and 4F subfamilies produce the majority of ω and ω-1 hydroxylated AA (known as AA hydroxylase) (Figure 3A) [76]. Both CYPs families are regio- and stereoselective [77,78]. For instance, CYP2C and 2J subfamilies can convert AA into four regioisomers of epoxyeicosatrienoic acids (EETs) depending on which double bond is involved in oxygen insertion, which results in 5,6-EET, 8,9-EET, 11,12-EET, and 14,15-EET. Each of the EET products can be either the R,S- or the S,R stereoisomer [77,78]. This regio- and the stereoselectivity of the epoxidation of AA to EET is CYP isoform-specific. For instance, while CYP2C8 in humans metabolizes AA with high regio/stereoselectivity to 14,15- and 11,12-EET (with ratio of 1.3:1 and more than 80% optical purity (OP) of R,S enantiomers), CYP2C9 shows very low regio- and stereoselectivity [79,80]. CYP2J2 is not regioselective as it produces all four regioisomers of EETs, in which 8,9- and 11,12-EET are largely a racemic mixture, and 14,15 -EET is mainly in the form of 14(R),15(S)-EET (OP ≈ 76%) [81]. CYP2C23, which is the main CYP involved in AA epoxidation in the rat kidney, generates 8,9-, 11,12-, and 14,15-EET in a ratio of 1:2:0.7 with high stereoselectivity (OP ≈ 95, 85, and 75%, respectively) [82]. Likewise, CYP2C44 in mice results in the same products [83]. The regioselectivity of different CYPs and the relative amount of each regioisomer are depicted in Figure 3C.

Even though the CYPs subfamilies involved in LA metabolism have not yet been meticulously examined, several studies proposed that all CYP isoforms can metabolize LA with high efficiency. For instance, CYP2C9 is known as the main LA monooxygenase in the human liver and generates both 9,10- and 12,13-epoxyoctadecamonoenic acids (EpOMEs) [84]. Other AA metabolizing CYP isoforms that can also accept LA as a substrate are CYP2C8 and -19; CYP2J2, -3, -5, and -9; CYP1A2; and CYP3A4 [85,86,87]. It is worth mentioning that the same CYP isoform can have different preferences as to whether they act as an epoxygenase or hydroxylase when the substrate is changed from AA to LA. For example, while human CYP2E1 has primarily hydroxylase activity on AA, it is a major LA monooxygenase [87].

Like LA, EPA and DHA have been shown to be substrates for CYP isoforms in human, rat, and mouse [22]. For example, the human isoforms CYP2C8, 9, 18, and 19, as well as CYP2J2 can epoxidize both EPA and DHA [88,89]. Moreover, the catalytic activities of CYP2C isoforms for EPA and DHA are almost the same as for AA, while CYP2J2 displays 9 and 2 times higher rates in metabolizing EPA and DHA, respectively, compared to AA [88,90]. Furthermore, these enzymes have different regioselectivity for EPA and DHA [88,90,91]. For instance, while CYP2C23 metabolizes AA to 8,9-, 11,12-, and 14,15-EET in a ratio of about 1:2:0.6, the epoxidation of EPA by CYP2C23 results in 17,18-, 14,15-, 11,12-, and 8,9- epoxy eicosatetraenoic acids (EEQs) in a ratio of about 6:1:1:1 [90,91]. Furthermore, while human CYP2C8 produces mainly 11,12- and 14,15-EET with AA as a substrate, it produces exclusively a terminal ω-3 PUFA epoxide with omega-3 PUFAs as a substrate, such as 17,18-EEQ with EPA, and 19,20- epoxy docosapentaenoic acid (19,20-EDPs) with DHA [88,90,91]. However, very low regioselectivity has been reported for other CYP2C isoforms toward EPA and DHA [22,89]. Furthermore, as previously mentioned, CYP2J2 has very low regioselectivity toward AA and high regioselectivity towards ω-3 PUFAs (preferentially generating terminal ω-3 PUFA epoxides) [88,90]. Furthermore, it should be noted that CYP2J2 and all CYP2C isoforms (except CYP2C8) exhibit high stereoselectivity, favoring the production of the R,S-enantiomers of 17,18-EEQ and 19,20-EDP [91,92].

**Figure 3 nutrients-12-03523-f003:**
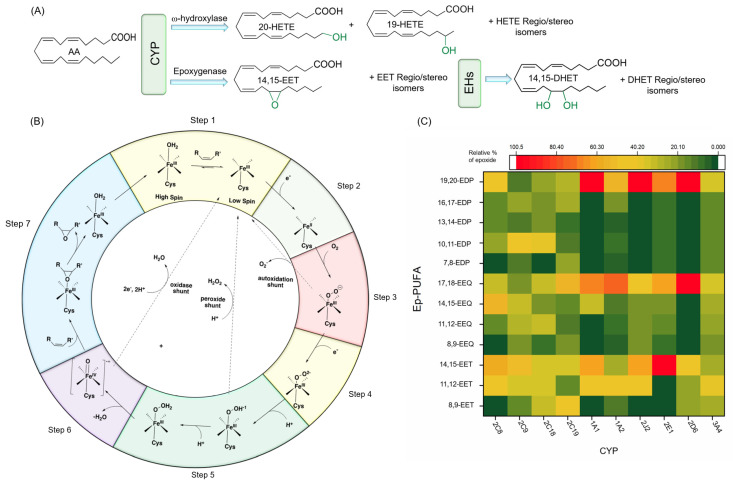
(**A**) CYP hydroxylase and epoxygenase activities on AA, as well as EH products of EETs. (**B**) CYP mechanism of function for monooxygenase activity. (**C**) Regioselectivity in epoxygenase activity of different CYP adapted from [88]. Note that all EETs and HETEs (except 20-HETE) can be exist as either the R- or S-enantiomers. Abbreviations- AA: arachidonic acid, CYP: cytochrome P450, DHET: dihydroxyeicosatrienoic acid, EDP: epoxydocosapentaenoic acid, EEQ: epoxyeicosatetraenoic acid, EET: epoxyeicosatrienoic acid, EH: epoxide hydrolase, HETE: hydroxyeicosatetraenoic acid.

#### 4.3.2. Regio- and Stereoselective Hydroxylation of PUFAs by CYPs

Like epoxidation activity, CYPs also hydroxylate in a regio-selective manner. CYP4A and 4F subfamilies hydroxylate AA at the terminal methyl group to produce 20-hydroxyeicosatetraenoic acid (20-HETE) as the major product, and 19-HETE as the minor product. Specifically, 20-and 19-HETE can be generated by human CYP4A11, rat CYP4A1, and mouse CYP4A12A, in a ratio of 90:10, 93:7, and 87:13, respectively [93,94,95]. CYP4F isoforms such as CYP4F2, CYP4F3A, and CYP4F3B are more regioselective than CYP4A in ω-hydroxylation [96]. On the other hand, the CYP1A1, CYP1A2, and CYP 2E1 mainly have ω-1 hydroxylase activity on AA, yielding 19-HETE as the predominant product, and 16-, 17-HETE (CYP1A1 and CYP1A2) and 18-HETE (CYP2E1) as minor products, while 20-HETE is not produced [22,76]. Interestingly, CYP2J9, unlike other members in the CYP2J subfamily, generates the 19-HETE from AA almost exclusively, while the other members in the subfamily such as human CYP2J2 and rat CYP2J3 mainly display epoxygenase activity towards AA [81,96,97]. CYP4A1, which is the main PUFA hydroxylase in rats, can metabolize LA at the same rate as AA and generates 18- and 17-hydroxy octadecadienoic acids (HODE) in a ratio of 3:1 [94]. Likewise, human CYP4A11 has shown hydroxylase activity on LA [87,98].

EPA and DHA can also be effectively hydroxylated by CYP hydroxylases such as human CYP4A11, CYP4F2, CYP4F3A, and CYP4F3B, as well as mouse CYP4A12, and rat CYP4A1 [95,96,99,100]. Again, when the substrate of these CYPs changes from AA to EPA or DHA, the regioselectivity, as well as reactivity, is altered. For instance, while CYP4A1 hydroxylates AA (generating mainly 20- and 19-HETE), it epoxidizes and hydroxylates EPA to produce predominantly 17,18-EEQ (68%) and 19-HEPE (31%) [94,100,101]. Furthermore, when DHA is a substrate, the CYP4A1 exclusively produces the epoxidized product: 19,20-EDP [88,92]. Likewise, CYP4A12A can function exclusively as ω/(ω-1)-hydroxylase with AA, producing 20- and 19-HETE in a ratio of 8:2, and can metabolize EPA exclusively to 17,18-EEQ with a minor amount of 20- and 19-HEPE [95]. This trend has also been observed for CYP2E1, as it acts as a hydroxylase to AA, and an epoxygenase to EPA and DHA [88,90]. Note that the change in catalytic preference from ω to (ω-1)-hydroxylase activity of CYPs has also been observed. For instance, by changing the substrate from AA to EPA and DHA the ratio of ω to (ω-1)-hydroxylase of CYP4A1 shifts from 4:1 to 1:3 and 1:2, respectively [96]. Considering CYP4F subfamilies, CYP4F3A and CYP4F3B display higher catalytic activity for AA and DHA compared to EPA, while CYP4F2 has a higher preference for hydroxylating DHA compared to AA and EPA [96]. CYP4F8 and CYP4F12 mainly function as (ω-n)-hydroxylases, with AA producing 18- and 19-HETE, and producing 17,18-EEQ and 19,20-EDP as the main products of EPA and DHA, respectively [102]. CYP2U1 is another CYP isoform that is expressed in the brain, which mainly acts as a ω-hydroxylase for ALA, AA, EPA, and DHA [103,104].

The regio- and stereoselectivity of CYP enzymes ultimately affects the physiology because numerous studies have suggested that the biological activity of Ep-PUFAs is controlled by the regio/stereoselectivity of the target receptors, which will be discussed in the following paragraphs (Section 4.4).

#### 4.3.3. Physiological Functions of Regio-/Stereoisomers of CYP Products: Ep-PUFAs and Hydroxy-PUFAs

The regio- and stereoselectivity of CYP enzymes ultimately could alter their effects on mammalian physiology because numerous studies have suggested that the biological activity of Ep-PUFAs is regio/stereoselective. These results will be briefly summarized in this section, whereas the specific effects of CYP PUFA metabolites on neurodegeneration will be discussed in Section 7.

One study regarding the regioselective effects of EET by Node et al., demonstrated that 11,12-EETs exert significant anti-inflammatory effects by inhibiting the TNF-α-induced vascular cell adhesion molecule-1 (VCAM-1) expression, while no effects were observed for 14,15-EETs [105]. In addition, peroxisome proliferator-activated receptor-α (PPARα), which plays an important role in inflammation, can only be activated by 8,9-EET, and 11,12-EET, but not 14,15-EET [106]. Besides, the omega-3 Ep-PUFAs also affect the regioselectivity of physiological processes. For example, EDPs alleviate nociception response in rodent inflammatory pain model with relative potencies of 13,14-EDPs > 16,17-EDPs > 19,20-EDPs [107]. The biological effects of Ep-PUFAs in mammals are also stereospecific. For instance, Ding et al. found that a G_s_-coupled receptor on the membrane of endothelial cells responds to 11(R),12(S)-EET, but not 11(S),12(R)-EET, which mediates protein kinase A (PKA)-dependent translocation and activation of transient receptor potential (TRP) C6 channels [108]. Over the last few decades, several studies have shown that the Ep-PUFAs likely act stereospecifically [100,109,110,111]. In addition to Ep-PUFAs, the biological functions of hydroxy-PUFAs are also stereo- and regioselective [22,112,113,114,115].

Overall, these studies show the significance of regio- and stereochemistry on the physiological effects of Ep- and hydroxy-PUFAs. One of the main challenges in investigating the regio-/stereoisomers of PUFA metabolites is obtaining significant quantities of pure regioisomers and their antipodes. Chemical epoxidation is neither chemoselective nor enantioselective, and the purification process of positional isomers and enantiomers is a tedious process [116]. While enzymatic epoxidation seems to be an effective way to generate single enantiomers, using a chemical inversion process is inevitable to access the corresponding enantiomers. Besides, this method is non-diversifiable and cannot be utilized in synthesize the corresponding analogs [116,117]. Progress in synthetic routes to achieve large quantities of specific regio-/stereoisomers with high purity is the key to improve our understanding of these isomers and the mechanisms of their functions.

### 4.4. CYP Enzymes in the Central Nervous System

The first evaluation of CYP in the brain was conducted in 1977 by Sasame et al. [118]. They found 30 pmol/mg of CYP enzymes in the rat brain, which was approximately 3% of the corresponding levels in the liver, with 30 times lower activity [118]. Since then, many efforts have been made to identify different CYP enzymes, their activity, as well as expression patterns within the brain. To date, 41 of 57 CYP enzymes have been identified in various brain regions [119]. Studies on CYP enzyme expression and function in the brain revealed some remarkable information. CYPs can be found in both glial cells and neurons either in the cell bodies or throughout the cell processes. For instance, isoforms such as CYP2E1, 1A1, 3A, and 2B, are mostly expressed in neurons, while others such as CYP2D6, are predominately expressed in both glial and neuron cells [120,121]. CYP enzyme expression in the brain is heterogeneous among different parts of the brain due to the presence of various cell types with different needs and functionality [119]. Furthermore, some CYP enzyme levels in specific neurons are even greater than their counterparts in hepatocytes. For example, CYP46A1, which regulates the cholesterol homeostasis in the brain, and CYP2D6, which is involved in the biosynthesis of serotonin and DA, are mainly expressed in the brain [122,123]. Even though CYP enzymes (especially 2C, 2J, and 4A) can potentially be highly influential in brain function through their monooxygenase activity in PUFAs metabolism (which will be discussed later), they are also key players in hormone, cholesterol, endocannabinoids, and neurotransmitter metabolism [18,124]. Therefore, these enzymes can affect neuronal activity and homeostasis through other mechanisms, independent of their PUFA metabolism (Figure 4). For instance, CYP2D6 expressed in the brain is involved in the metabolism of endogenous neural compounds such as catecholamines and can metabolize drugs and inactivate neurotoxins such as 1-methyl-4-phenyl-1,2,3,6-tet-rahydropyridine (MPTP) and 1-methyl-4-phenylpyridinium (MPP^+^). Thus, low activity of this enzyme can result in neuronal hypofunction, especially in dopaminergic neurons. Mann et al. demonstrated that CYP2D6 levels increase with age; however, in PD patients, this enzyme is expressed 40% less compared to in a healthy brain [125]. Lower levels of this enzyme in PD patients may reduce their ability to inactivate PD-causing neurotoxins. Furthermore, CYP46 and CYP27 seem to be important in AD and Huntington disease, probably due to their role in the brain cholesterol homeostasis and metabolism [126,127,128]. In addition, genetic variations in CYP19 and CYP2J2 have been associated with enhanced susceptibility AD [129,130]. The rs890293 variant of CYP2J2, which results in a CYP2J2 enzyme with reduced function, has been shown to be associated with late-onset AD in the Chinese Han population [131]. These data suggest a possible role of Ep-PUFAs, dihydroxy-PUFAs, or hydroxy-PUFAs in neuroprotection and reduced risk of AD, as CYP2J2 is one of the key isoforms of CYP responsible for Ep-PUFAs metabolism. There are extensive studies on the CYP enzymes effects in neuronal function and homeostasis, which is beyond the scope of this paper, and we are only focusing on the role they have in Ep-PUFAs or dihydroxy-PUFAs metabolism.

## 5. EH: Epoxy Hydrolase, a Critical Member in Ep-PUFA Metabolism

The epoxide is a three-membered heterocycle that has unfavorable bond angles and a polarized C-O bond leading to significant electrophilic activity [132]. Thus, epoxides could covalently react with a nucleophile and cause a wide range of biological and pathological effects. For instance, styrene epoxide derivatives can be attacked by nucleophilic exocyclic amino groups of nucleotides or N7 moiety of purines causing DNA adducts and mutations [133,134]. There is extensive evidence confirming the potential of some epoxides, particularly Ep-PUFAs, to act as secondary messengers in the initiation of different physiological pathways. Epoxide hydrolases (EHs) catalyze the hydrolysis of both endogenous and exogenous epoxides, resulting in corresponding 1,2-diol compounds [135]. Therefore, EHs are involved in detoxification and regulating signaling molecule metabolism by hydrolyzing epoxides and modulating their endogenous levels.

The EHs can be detected in both prokaryotes and eukaryotes. There are seven different EHs identified in mammals (i) microsomal epoxide hydrolase (mEH, encoded by *EPHX1*), (ii) soluble epoxide hydrolase (sEH, encoded by *EPHX2*), (iii) epoxide hydrolase 3 (EH3, encoded by *EPHX3*), (iv) epoxide hydrolase 4 (EH4, encoded by *EPHX4*), (v) hepoxilin hydrolase, (vi) leukotriene A4 (LTA4) hydrolase, and (vii) cholesterol epoxide hydrolase [135,136,137,138,139,140]. The mEH, sEH, EH3, and EH4 enzymes can be considered as EH subfamilies, which are members of ⍺/β-fold hydrolases superfamily, comprising of eight anti-parallel β-strands as the core domain connected together by ⍺-helices that are interrupted by an adjustable lid domain. The other three enzymes can be categorized in different families due to their different catalytic mechanisms and substrate preferences that are well reviewed elsewhere and will not be discussed further [136,137,139]. Note that paternally expressed gene 1 (*Peg1*)/mesoderm-specific transcript (*Mest*) gene (*Peg1/Mest*) is another candidate that might be considered an EH due to its considerable sequence similarity to ⍺/β hydrolases. However, its enzymatic function is currently unknown [141]. Thus, we are using EH to refer to mEH, sEH, EH3, and EH4 in this review, unless otherwise mentioned. In order to avoid confusion, when we are refering to the genes for EHs, we will use the abbreviation *EPHX* (for humans; *Ephx* for rodents and other non-domesticated animals).

### 5.1. Characteristics of the Main EH Enzymes

Among the EHs, mEH is the first identified mammalian EH, which is encoded by the *EPHX1* gene and has a primary structure of 455 amino acids [142,143]. This membrane-bound enzyme is attached to the surface of the ER or the plasma membrane by its N-terminal membrane anchor [144,145]. The mEH also has an N-terminal extension that wraps around the protein and holds the lid domain down to the α/β hydrolase fold [146,147]. The localization of N-terminal in the membrane is a mechanism by which the C-terminal region with epoxide hydrolase activity faces the cytosol on ER membranes, and on the plasma membrane it is exposed to the extracellular medium. The mEH and CYP enzymes are type I membrane-bound proteins, and there is evidence of close proximity of CYP and mEH in the endoplasmic reticulum [148], suggesting a possible physical interaction. Interestingly, when mEH dissociates from the membrane and is found in the blood, this is considered a preneoplastic antigen, a marker for tissue damage and cancer [149]. sEH, on the other hand, can selectively hydrolyze lipid epoxides with a high catalytic rate, while having an unknown role in xenobiotic metabolism in the liver [140]. Human sEH, encoded by *EPHX2*, is a 62 kDa homodimeric enzyme located in the intracellular environment (cytosol and peroxisomes) [140]. Each monomer has two regions: a C-terminal region with epoxide hydroxylase activity, and an N-terminal region with phosphatase activity, which are linked together by a proline-rich linker (Figure 5A) [150]. Both mEH and sEH are widely found in different tissues with the highest levels in the liver, and their expression as well as specific activity can be altered by tissue, sex, and age [140,151].

Both EH3 and EH4 are predicted to be single-pass type II membrane proteins and possess N-terminal membrane anchors based on their amino acid sequences, yet these properties need to be experimentally confirmed. EH3 and EH4 have 45% homology in their sequence and were originally named as α/β hydrolase domain containing protein 9 (ABHD 9) and protein 7 (ABHD 7), but were renamed after studies done by Arand’s group showed their epoxide hydrolase activity toward epoxy octadecenoic acids (EpOMEs) and EETs [153,154]. However, their specific in vivo function is still under investigation. EH3 is a 41 kDa (360 residues) protein encoded by the *EPHX3* gene, and its microsomal properties have been identified by a related gene in insect cells [154]. EH3 expression is generally low compared to mEH and sEH. Isolated mRNA from a representative set of mouse organs revealed the lowest expression in the liver and kidney, while the highest expression was observed in the skin, stomach, lung, and tongue [154]. This expression pattern suggests a potential function of EH3 in barrier formation, which is supported by a recent study showing high turnover hydrolyzing activity toward a skin-related epoxide involved in a key step of water permeability barrier formation in the outer epidermis [155]. EH3 is also involved in leukotoxin metabolism to mediate acute respiratory distress syndrome (ARDS) [156]. In addition, several studies identified the epigenetic silencing of EH3 in different types of cancers. For instance, hypermethylation of EH3 is associated with prostate cancer relapse [157], and has been observed in human colorectal carcinomas [158], gastric cancers [159], as well as malignant melanomas [160].

EH4 is a 42 kDa (362 residue) protein encoded by the *EPHX4* gene located on chromosome 1p22.1. As already mentioned, like mEH and sEH, both EH3 and EH4 possess an aspartate (Asp) nucleophile, distinguishing them from other ABHD proteins that have either a serine (Ser) or Cys as nucleophile residue in the catalytic triad, placing them in the EH family [153,154]. EH4 expression is much higher in the brain compared to other tissues [141]. EH4 has been recognized as a target gene for the zinc finger protein 217, an important oncogene that is amplified and overexpressed in a variety of human tumors [161]. In addition, hypermethylated EH4 has been found in human colorectal cancer. Thus, EH4 is associated with the pathogenesis of some cancers [162]. In general, very little is known about the EH4 substrate spectrum and enzymatic function. However, the presence of a nucleophilic Asp in the catalytic triad (that distinguishes EH3 and 4 from other ABHDs), and high homology with two EHs in *C. elegans* [17], suggest the potential for EH4 to be an active EH with important physiological activity in the brain [153].

### 5.2. Catalytic Function and Mechanism of EHs

The major function of EH is to hydrolyze xenobiotic epoxides and Ep-PUFAs (Figure 3A). The catalytic activity of EHs can be described in three steps, with the contribution of four amino acids, which are involved in transforming the epoxide to a diol [135,163]. In the first step, the epoxide enters through the L-shaped hydrophobic tunnel with the nucleophilic aspartate in the center, and it is stabilized there by van der Walls and hydrogen bonding with the hydrophobic pocket and two tyrosines, respectively. The second step is the formation of an ester intermediate, in which the nucleophilic aspartate and polarizing tyrosine are involved. In this step, the oxygen of the epoxide is polarized by hydrogen bonds between the hydroxyl group of two tyrosine residues. At the same time, the carboxylate of Asp (a nucleophilic amino acid), which is opposite of the tyrosine residues, gets orientated and activated by other amino acids in the active site including histidine and an acidic residue (aspartate in sEH and glutamate in mEH) adjacent to the nucleophilic aspartate. Then, the activated nucleophilic aspartate attacks the electrophilic carbon of the epoxide resulting in a hydroxyl acylated-enzyme intermediate. The third step is diol formation. When the intermediate is formed in the second step, the His orients and then the charge relay system between the glutamate/aspartate and His activates the His to facilitate hydrolysis of the acylated intermediate, forming the corresponding 1,2-diol product. Table 1 shows the different residues in the catalytic triad of mEH, sEH, EH3, and EH4. Besides these highly conserved regions, the amino acid sequences and the N-terminal region are largely different among different EHs, probably due to their origin in evolutionary pathways.

### 5.3. Major EHs Responsible for Ep-PUFA Metabolism

sEH is the main enzyme that degrades Ep-PUFAs, but other EHs also play a role in Ep-PUFA metabolism [85]. mEH selectively hydrolyzes arene and cyclic epoxides, is involved in xenobiotic metabolism, and catalyzes the epoxidation of a wide and still growing range of substrates [151]. The expression of mEH in specific neuronal populations suggests a previously unnoticed role in processing endogenous substrates. Zeldin et al. reported that mEH hydrolyzes EET with up to an 8-fold lower K_m_ than sEH [164]. However, the catalytic efficacy (k_cat_/K_m_) of mEH is still five- to 50-fold lower than that of sEH for all EET regioisomers. The regio-selectivity of the two enzymes, based on the catalytic efficacies, was 11,12-EET > 8,9-EET = 14,15-EET >> 5,6-EET for mEH and 14,15-EET = 11,12-EET > 8,9-EET > 5,6-EET for sEH [165].

Considering the role of EH3 in the Ep-PUFA metabolism, an in vitro study using EETs as substrates demonstrated that EH3 has high catalytic efficacy (similar to sEH), and the highest specific activity reported thus far among EHs. These data suggest that even low expression of EH3 would be enough for a high concentration of EETs produced in cells [154]. On the other hand, the substrate spectrum of EH4 is unknown, mainly because it has not shown enzymatic activity on any of the substrates tested so far. The lack of turnover could be due to the expression of a misfolded protein, because EH4 is potentially membrane bound, complicating recombinant expression. Therefore, activity is still speculative in the case of EH4.

### 5.4. EHs in the Brain

In this section, the different EHs and their distribution along with the potential application are discussed.

#### 5.4.1. mEH in the Brain

In the brain, mEH is highly expressed in epithelial cells, especially in cerebral blood vessels and the choroid plexus that form the blood–brain barrier, where mEH might be mainly involved in detoxification and protecting CNS. This enzyme is also found in smooth muscle cells and specific neuronal cells including, but not limited to, cerebellar granule cells, central amygdala neurons, striatal neurons, hippocampal pyramidal neurons, and in some astrocytes [151,166,167]. Figure 5B shows the distribution of EHs in different regions of the human brain using mRNA expression techniques [152]. The role of mEH in neurological disorders is a subject of intense interest in the field of pathophysiology-related mEH due to its high expression in the brain [151]. In support of this, Lui et al. identified expression of mEH in the hippocampus and in the cortex of AD patients [168]. They also found that exposure to β-amyloid aggregation could induce mEH expression in astrocytes in primary hippocampal glial culture. Overexpression of mEH has also been observed in the rat hippocampus and entorhinal cortex when the animals were treated with trimethyl-tin, an environmental neurotoxic agent, further supporting the role of mEH in the pathogenesis of neurodegeneration and neurotoxicity [168]. Genetic ablation of mEH in mice treated with methamphetamine caused a decrease in synaptosomal DA uptake in the striatum, while enhancing the extracellular DA levels in the nucleus accumbens compared to the wild-type mice, suggesting that mEH is a potential therapeutic target for drug addiction treatment [169]. In addition, increased levels of mEH have been found in human brain tumor cells, further supporting the pivotal role of mEH in brain-related pathogenesis [170].

Although the role of mEH xenobiotic metabolism has been extensively studied, its distribution suggests a possible endogenous role [171,172]. Even though mEH possesses low catalytic activity towards Ep-PUFAs, high levels of mEH expression in specific brain regions and neurons, as well as evidence for substrate channeling between mEH and CYP on ER, suggests that mEH contributes to Ep-PUFA hydrolysis in the brain [166,173,174]. For instance, tracking of dihydroxyeicosatrienoic acids (DHET) regioisomers formation in freshly isolated mouse brain hippocampal cells of WT, mEH^−/−^, and sEH^−/−^ mice revealed substantial mEH-dependent EET turnover when cells were treated with AA. Furthermore, hippocampal cells treated with 1-adamantyl-3-cyclohexylurea (ACU), a selective sEH inhibitor, could still hydrolyze EETs, further supporting a possible role of mEH in the brain Ep-PUFAs metabolism [165]. In short, while mEH activity in hydrolyzing Ep-PUFAs is well studied, its role in neuronal signaling and hemostasis is mostly unknown.

#### 5.4.2. sEH in the Brain

sEH is expressed in different brain regions (Figure 5B) and in multiple cell types such as astrocytes, endothelial cells, oligodendrocytes, and neural cell bodies [152,173,174]. sEH expression in microglia cells has not yet been well characterized; however, Huang et al. found sEH expressed in the BV2 microglial murine cell line [175]. They also found that genetic ablation or pharmaceutical inhibition of sEH can attenuate microglia activation, suggesting a significant role in neuroinflammation [175]. Consistent with this study, researchers have revealed a role for sEH in different neuroinflammatory and neurodegenerative pathogenesis involving microglia, astrocytes, and neurons by its contribution to Ep-PUFA metabolism [21,176,177], which will be discussed later. Furthermore, sEH is critical to pathways related to axonal growth in neurons as well as neuronal development, mainly through its Ep-PUFA hydrolyzing activity [178]. Taken together, these data point to sEH as a potential therapeutic target.

#### 5.4.3. EH3 and Other EHs in the Brain

Expression of EH3 is low in almost all tissues, including the brain. However, the stomach, skin, and lung are organs with the highest EH3 levels, suggesting that EH3 might be mostly involved in barrier formation [154,155]. Hoopes et al. used *Ephx3* knockout mice to track EH3 function in vivo [179]. Interestingly, they found no difference in endogenous epoxide:diol ratios (such as EETs:DHETs or EpOMEs:dihydroxyoctadecamonoenoic acids (DiHOMEs) ratios) in different tissues compared to wild-type tissues. In addition, *Ephx3* ablation had no effect on the mRNA levels of CYP epoxygenase, and the *Ephx*^−/−^ mice exhibited a similar inflammatory response to lipopolysaccharides (LPS) as compared to wild-type mice. Furthermore, no overt phenotype, i.e., no significant change in weight, and anatomical normality of organs, and reproductive capacity, was observed in the *Ephx3*^−/−^ mice [179]. Therefore, sEH and mEH are predominant enzymes involved in diol formation. In contrast to *Ephx3*, *Ephx4* is expressed mainly in the brain, with the highest expression in the neocortex and hippocampus (Figure 5B) [152]. Similar to *Ephx3*, there is very low expression of *Ephx4* in the liver and intestine [180].

### 5.5. Possible Physiological Roles of Dihydroxy-PUFAs

Dihydroxy-PUFAs generated by EHs possess higher polarity compared to their Ep-PUFA precursors. Thus, they tend to reside in the extracellular fluid, and are generally considered to have little to no activity [23]. Despite this, several studies suggest a possible physiological function of the PUFA diols generated by EHs. For instance, an early study from Moghaddam et al. showed that dihydroxy octadecenoate (DiHOME), an EH product of EpOME [156]. In addition, Weintraub et al. demonstrated that preincubation of 14,15-DHET, and 11,12-DHET can augment both the magnitude and duration of inflammation-induced relaxation in pig porcine coronary arteries [181]. Oltman et al. showed that the DHET regioisomers were as potent as (or, in the case of 11,12-DHET, more potent than) their parent EETs in vasodilation of dog canine coronary arterioles [182]. Larsen et al. also found that DHETs can modulate vasomotor tone in the resistance vessels of the human heart and can induce vasodilation via a BK_Ca_ channel-mediated mechanism [183]. In addition, Abukhashim et al. observed that while 11,12-EETs can modulate cAMP production, the corresponding 11,12-DHET metabolite can function as a negative controller to limit cAMP production in HEK293 cells [184]. Another study found 14,15-DHET can activate PPARα [185,186]. Sisemor et al. found that DiHOMEs can disrupt mitochondrial function, specifically through activation of the mitochondrial permeability transition [187]. Consistent with this, Moran et al. showed an inhibitory effect of DiHOMEs on the mitochondria respiratory chain in a regioselective manner, as 12,13-DiHOME is approximately 4-fold more potent than 9,10-DiHOME [188]. A study by Stanford et al. also demonstrated that 12,13-DiHOME, a lipokine produced by brown adipose tissue, increases fatty acid uptake in muscle [189]. Interestingly, Kundu et al. also reported that DHETs are essential to monocyte chemoattractant protein-1 (MCP-1) dependent chemotaxis, suggesting a pro-inflammatory role [190]. Furthermore, various studies showed the presence of dihydroxy-PUFA in human urine or plasma, and these levels correspond to various diseases. For instance, glucuronic acid conjugates of DiHOME were found in the urine of children with generalized peroxisomal disorders. Furthermore, blood serum analyses of AD patients showed about 20% higher levels of all four DHET species compared to healthy individuals [20]. Furthermore, individuals with type 2 diabetes and AD showed higher levels of 14,15-dihydroxy eicosatetraenoic acid (14,15-DiHETE) (66%), and 17, 18-DiHETE (29%) compared to type 2 diabetes patients with healthy cognitive function, with no difference in DHET species between these two groups [20]. These studies suggest the potential biological effects of dihydroxy-PUFAs, which need to be elucidated.

## 6. CYP PUFA Metabolism and the Nervous System

The brain has a high capacity for the *de novo* synthesis of Ep-PUFAs from their parent PUFAs by CYP enzymes. In neuronal cells, the resulting Ep-PUFAs are rapidly degraded by EHs such as sEH [191,192]. While the CYP pathway seems to be the dominant route of the production of Ep-PUFAs in the brain, they may also be transported to the brain by cellular uptake [193,194]. The major elimination pathway of Ep-PUFAs is their hydrolysis by EHs to more polar 1,2-diols, whereas spontaneous hydration, beta-oxidation, chain elongation, and reincorporation into the phospholipids are also potential mechanisms to maintain the homeostasis of Ep-PUFAs [192]. Note that each of the Ep-PUFA metabolic pathways can have their own biological functions, which has yet to be fully understood [190]. For example, incorporation of Ep-PUFAs into lipid bilayers might not eliminate their biological activity, as studies demonstrated the activity of membrane-bound Ep-PUFAs such as EETs [194]. Furthermore, the regioselectivity of both the CYP and sEH might potentially modulate the concentration of specific Ep-PUFAs in tissues and brain. For instance, the concentration of 7,8-epoxydocosapentaenoic acid (EDP), which is not a preferred substrate of sEH, is almost 30 times higher than other regioisomers in the brain and spinal cord of rats [107]. Ep-PUFAs can also be further oxidized or bind to fatty acid binding protein (FABP) [192].

Besides, several studies demonstrated that the brain oxylipins might be amenable to the dietary lipid content [14]. For instance, ω-3 PUFA (EPA and DHA) supplementation enhances the level of EPA-derived and DHA-derived CYP metabolites and decreases AA-derived CYP metabolites in the brain of mice [195,196]. Ostermann et al. found that a diet enriched with ω-3 PUFA EPA and DHA reduced Ep-PUFA level and the ratio of epoxy- to dihydroxy-PUFA in murine brain, revealing higher sEH activity [197]. Rey et al. also observed that dietary ω-3 PUFAs supplementation promotes the synthesis of pro-resolving oxylipins in the mouse hippocampus [198]. Moreover, feeding rats with ω-6 PUFA (LA) increased the endogenous level of oxylipins derived from AA and LA and reduced EPA-derived oxylipins in the brain cerebral cortex [199], while LA deficient diet decreased LPS-induced pro-inflammatory PGE2 formation in rat brains [200]. Note, that even though the dietary PUFAs effect on brain FAs composition has been well-studied, oxylipin profiles do not necessarily reflect tissue precursor PUFA composition [201,202,203]. For example, the high DHA to AA ratio in the CNS system does not necessarily lead to high levels of DHA-derived oxylipins in the CNS.

## 7. Ep-PUFAs and Neuroinflammation

Neuroinflammation is a complex process required for homeostasis maintenance in the CNS including clearance of cellular debris, β-amyloid plaques, and glial scars that if unmitigated can result in chronic inflammation, leading to neural pathogenesis [204,205]. Neuroinflammation also contributes to neurodevelopment, immune conditioning against infections, as well as clearance of damaged tissues upon injury [204]. Microglia and astrocytes are the main cells involved in the neuroinflammatory response [206]. Activated microglia produce pro- and anti-inflammatory cytokines such as interleukins 1β and 6 (IL-1β and IL-6), tumor necrosis factor alpha (TNFα), interferons α and γ (IFN-α and IFN-γ), chemokines like IL-8, and macrophage inflammatory proteins 1α and 1β (MIP-1α, MIP-1β), neurotrophins such as brain-derived neurotrophic factor (BDNF) and nerve growth factor (NGF), growth factors such as fibroblast growth factors (FGFs) and transforming growth factor β (TGF-β), ROS and RNS, inflammatory markers such as serum amyloid P and C-reactive protein, along with proteases and complement system proteins [204,206,207]. Perivascular macrophages and endothelial cells also play a significant role in the interpretation and propagation of inflammatory signals within the CNS. Even though the neuroinflammatory response aspires to remove an injury or insult, when resolution fails, it becomes a chronic inflammation which is related to numerous major disease states [26]. Thus, neuroinflammation can cause or exacerbate the pathogenesis of NDs such as the PD, AD, depression, etc. [26,205,208]. In this section, we focus on recent discoveries about the role(s) of Ep-PUFAs, which regulate acute and chronic neuroinflammation.

### 7.1. Acute Neuroinflammation

A promising therapeutic potential of Ep-PUFAs is mitigating neuroinflammation caused by acute nervous system damages such as seizures, brain trauma, and ischemic and hemorrhagic strokes [206,209,210]. The neural damage/loss resulting from these insults activates the resident immune cells, including microglia and astrocytes, to repair the injured tissues. However, an uncontrolled activation can result in chronic damage, as observed in other organ systems [26,205,208]. The neuroinflammation can be controlled by either suppressing the expression and release of pro-inflammatory mediators such as iNOS, Iba1, TNF-a, or increasing anti-inflammatory cytokines, including IL-10 by polarized microglia, which is greatly affected by endogenous Ep-PUFAs, in particular EETs [136,138,139]. Several studies have shown that deletion or inhibition of sEH induces production of anti-inflammatory cytokines such as IL-10, while reducing pro-inflammatory mediators including IFN-γ, and decreasing TNF-α localization and circulation [176,211,212].

Several studies in chemical-induced seizures murine model suggested that Ep-PUFAs could suppress neuroinflammatory. Inceoglu et al. showed that genetic knockout or pharmaceutical inhibition of sEH attenuated seizures induced by GABA receptor antagonists such as picrotoxin and pentylenetetrazole [16]. However, little to no effect was observed with sEH inhibition in cases where seizures were induced by 4-aminopyridine, a potassium channel blocker. Vito et al. reported that coadministration of an sEH inhibitor and diazepam could protect against the progression of tonic seizures and lethality induced by tetramethylenedisulfotetramine in a mouse model [213]. In addition, Huang et al. observed increased levels of both sEH and pro-inflammatory cytokines, including IL-1β and IL-6, in the hippocampus of mice that underwent pilocarpine-induced seizures, further suggesting a potential relationship between sEH and seizures [214]. Using double-immunofluorescence labeling, they indicated that astrocytes are the major source of sEH in the hippocampus (subfields of the dentate gyrus, CA1, CA3, and dentate gyrus). This result suggested that alternation of sEH in the hippocampus may contribute to the observed significant astrogliosis in response to seizures. Finally, it has been shown that inhibition of sEH suppresses pilocarpine-induced seizures and increases the seizure-induction threshold [135]. Similarly, it was reported that deletion or inhibition of sEH suppress neuro-inflammation caused by cortical impact injury (a model of traumatic brain injury (TBI)) [175].

TBI leads to an induction of CYPs, and increased levels of PUFAs (especially AA and DHA), leading to a dynamic change of Ep-PUFA and an upregulation of sEH in the injured brain [215]. Hung et al. found that 12-[[(tricyclo[3.3.1.13,7]dec-1-ylamino)carbonyl]amino]-dodecanoic acid (AUDA, an sEH inhibitor) treated mice had higher levels of EETs and an increase of EET:DHET ratio in the injured brain suppressing inflammatory responses [175]. *Ephx2* deletion in mice was shown to reduce blood-brain barrier (BBB) permeability, brain edema, neural apoptosis, and death. These changes were associated with significantly reduced EET degradation, inflammatory mediator expression, microglial/macrophage activation, and IFN-γ-induced nitric oxide (NO) production. Furthermore, a treatment of an sEH inhibitor in primary microglial cultures attenuated LPS- or IFN-γ-stimulated NO production, and decreased LPS- or IFN-γ-induced p38 MAPK and NF-κB signaling. These effects were abolished by co-administration with a 14,15-EET antagonist (14,15-EEZE), suggesting that 14,15-EET could be a major mediator of the sEH inhibition [175]. Several studies have shown that 14,15-EET can induce secretion of vascular endothelial growth factor (VEGF) and BDNF to protect the existing neurons during inflammation [176,216]. Apart from the protective effects of EETs through their actions on auxiliary cells, EETs can directly impact neurons, for instance, by enhancing their neurite outgrowth [123,143,178,217]. In addition, both genetic ablation and pharmacological inhibition of sEH could attenuate the functional and historical deficits in cerebral ischemia by vascular and neural protection [218,219]. These effects suggest that sEH inhibition and increased levels of Ep-PUFAs may reduce CNS injury both by reducing the inflammatory response in resident immune cells and by direct neuroprotective effects on the neurons through modulation of the endogenous level of beneficial Ep-PUFAs.

### 7.2. Chronic Neuroinflammation

The acute neuroinflammatory response caused by glial cell activation leads to repair of damaged areas of the brain and spinal cord. However, persistent and dysregulated neuroinflammatory response may shift acute neuroinflammation to chronic neuroinflammation [26,205,208]. Chronic activation of glial cells, which leads to a chronic neuroinflammatory state, protein accumulation, ER stress, mitochondria dysfunction, uncontrolled oxidative stress, axonal transport impairment, and apoptosis. Thus, it causes detrimental impacts on neuronal function and can lead to NDs, such as AD, PD, amyotrophic lateral sclerosis, and neuropsychiatric disorders including schizophrenia and depression [26,27].

#### 7.2.1. Alzheimer’s Disease

AD is a gradual degenerative brain disease, mostly involving the hippocampus and neocortex, which are associated with progressive memory disorders and cognitive dysfunction, as well as psychiatric symptoms [3,220,221]. Two main pathological characteristics of AD are aggregates of tau protein inside neurons and extracellular senile plaques due to depositions of amyloid-β (Aβ) protein fibrils [222]. Moreover, the Aβ in soluble form is extensively studied as one of the major causes of the pathogenesis of AD, primarily by inducing mitochondrial dysfunction [223,224]. It has been shown that the blocking endogenous EET production by a selective CYP epoxygenase inhibitor, N-(methylsulfonyl)-2-(2-propynyloxy)-benzenehexanamide (MS-PPOH), aggravated the effect of Aβ in astrocyte mitochondrial dysfunction, whereas pretreatment of primary hippocampal astrocyte culture with exogenous 11,12-EET or 14,15-EET prevented Aβ-induced mitochondrial depolarization and fragmentation [225]. In addition, Sarkar et al. found that Aβ causes a significant decrease in the level of both DHETs and EETs in microsomes isolated from the rat cerebral cortex in a region- and cell-specific manner, which could be due to a decrease in CYP activity after Aβ exposure [226]. However, another study demonstrated an increased level of AA-derived EETs in the hippocampus of human amyloid precursor protein (hAPP) expressing mice, suggesting increased activity of CYP or a decreased level of sEH [227]. Furthermore, several studies showed an increase in the level of DHET in AD patients. For instance, a blood serum analysis of the AD subjects showed higher levels of all four DHET species: 5,6- DHET (15% higher), 8,9-DHET (23% higher), 11,12 DHET (18% higher), and 14,15-DHET (18% higher) compared to healthy individuals [20]. Moreover, sEH is upregulated in multiple transgenic AD mouse models as well as in human AD brains in many studies [168,228]. Thus, the contradictory results regarding the EET and DHET levels may be due to the difference in experimental design and the models used.

In a further effort to study the effects of EHs in AD, Lee et al. found upregulation of sEH in the brain and predominantly in hippocampal astrocytes in a murine model of early-onset AD (*APP/PS1 Tg*) with severe AD-impaired pathology [229]. Genetic deletion of *Ephx2* (sEH gene) in APP/PS1 Tg mice decreased Aβ plaque formation, increasing astrogliosis in the brain, higher production of anti-inflammatory cytokines, and increased activity of two different transcription factors, nuclear factor kappa-light-chain-enhancer of activated B cells (NF-κB) and nuclear factor of activated T cells (NFAT). The *APP/PS1 Tg/Ephx2^−∕^**^−^* mice also had improved memory formation, spatial learning, and nesting building ability [229]. These data suggest a pivotal role of sEH in the regulation of neuroinflammation in AD pathology. A study by Gosh et al. also found a significant upregulation of sEH in a β-amyloid mouse model (5xFAD) and in postmortem human AD brain samples. The oxylipin analysis in transgenic 5xFAD mice showed a drastic reduction in Ep-PUFAs, in particular EDP and EETs. Orally administrated N-[1-(1-oxopropyl)-4-piperidinyl]-N’-[4-(trifluoromethoxy)phenyl)-urea (TPPU), a potent sEH inhibitor, in the 5xFAD mice reinstated the Ep-PUFA levels and reversed astrocyte and microglia reactivity as well as immune pathway dysregulation [228]. Consistent with this result, another study examined administration of three structural different sEH inhibitors in two AD mouse models (5XFAD and SAMP8, paradigms of early-onset and late-onset AD) [230]. They found a reduction in gene expression and brain protein levels of pro-inflammatory cytokines, including tumor necrosis factor–α (TNF-α), IL-1β, C-C motif ligand 3(CCL3), in both mouse models. They also observed an overall decrease in ER stress and oxidative stress when AD mouse models were treated with inhibitors, though the magnitude of the effect was different among the inhibitors [230]. These data reveal that the biological outcomes observed were not due to off-target effects related to a specific sEH inhibitor. These studies could well support the potential therapeutic effects of sEH inhibition in regulating neuroinflammation, and reducing tau hyperphosphorylation pathology, amyloid plaques formation, ER stress, oxidative stress, and cognitive impairment. However, the detailed mechanism underlying sEH-mediated regulation of AD pathologies such as Aβ formation and clearance demands more investigation.

While the role of mEH in AD is still unknown, the level of mEH expression is likely regulated during brain insults, causing a significant upregulation of the enzyme. For instance, a high level of *mEH* expression is found in activated astrocytes in epileptic tissue and around β-amyloid plaques in the brain tissue of patients with AD symptoms [168].

#### 7.2.2. Parkinson Disease

Parkinson’s disease (PD) is the second most frequent neurodegenerative disease after AD [231]. Although the precise pathogenesis of PD remains unknown, loss or dysfunction of dopaminergic neurons in the substantia nigra (SN) pars compacta (SNpc) and deposition of accumulation of misfolded α-synuclein in intra-cytoplasmic inclusions called Lewy bodies (LBs) are considered as hallmarks of PD [231,232]. In addition, enhanced levels of inflammatory mediators including IL-1β, IFN-γ, and TNF-α are evident in various studies with PD models [231,232]. To date, there are no pharmaceutical curative treatments approved by the Food and Drug Administration (FDA) [232].

MPTP (1-methyl-4-phenyl-1,2,3,6-tetrahydropyridine) is a precursor compound for the mitochondrial complex I inhibitor 1-methyl-4-phenylpyridinium (MPP^+^) that induces neurotoxicity including dopaminergic loss or dysfunction (such as loss of tyrosine hydrolase-positive (TH^+^) cells, loss of DA transporter (DAT), and increased oxidative and ER stress. Several studies also confirmed that overexpression of sEH in the striatum is accompanied by MPTP treatment, while sEH ablation protects neurons against MPTP-induced neurotoxicity in the mouse striatum. Qin et al. showed that administration of MPTP to mice causes a dramatic loss in the TH^+^ neurons, while sEH is upregulated in the substantia nigra. Interestingly, genetic knockout of sEH or treatment with an sEH inhibitor (and to a lesser extend 14,15-EET), attenuated the neurotoxicity induced by MPTP [233]. Ren et al. also demonstrated that MPTP-induced neurotoxicity in the striatum and substantia nigra was attenuated after subsequent repeated oral administration of TPPU [19]. Furthermore, an *in vitro* study found that TPPU has a therapeutic effect directly on induced pluripotent stem cells (iPSCs) with a parkin RBR E3 ubiquitin protein ligase (*PARK2*) mutation, by suppressing the increase in caspase-3 cleavage, thereby reducing apoptosis in dopaminergic neurons [19]. In addition to these studies, several other studies have shown that the sEH expression positively correlates with phosphorylation of α-synuclein in the striatum, which further supports a role for sEH and Ep-PUFAs in the pathogenesis of neurological disorders such as PD and shows promising potential for sEH as a biomarker for PD diagnosis [234,235]

### 7.3. Potential Molecular Targets of Ep-PUFAs in the Nervous System

Ep-PUFAs such as EETs are involved in inflammation through multiple mechanisms as demonstrated in animal models and tissues. However, no specific receptors for Ep-PUFAs have been identified yet. Therefore, a mechanistic study of Ep-PUFAs is difficult. Over the years, it has been shown that EETs initiate their anti-inflammatory effects through NF-κB, which is a transcription factor with significant roles in cell survival, immune responses, and inflammation. Several studies showed that increased activity of CYP2J2 and/or EETs could reduce inflammation by suppressing the degradation of IκBα, an endogenous NF-κB inhibitor, thus reducing the activation of NF-κB and inflammation. For instance, NF-α induced nuclear translocation of NF-κB in endothelial cells by preventing IκBα degradation through inhibition of IκB kinase (IKK) activity [105]. However, IKK and IκBα regulation is intricate, and more research is needed to elucidate how EETs and other EpFAs regulate the activity of NF-κB. Decades of research indicate that the mechanism of action of EETs and other Ep-PUFAs is through a direct interaction with peroxisome proliferator-activated receptor (PPAR), a putative membrane-bound G protein-coupled receptor (GPCR), and ion channels such as transient receptor potential (TRP) channels (Figure 6A) [174,194]. Below, we will discuss several potential targets for Ep-PUFAs in regulating neuronal function and neuroinflammation.

#### 7.3.1. PPARs

PPARs are involved in regulating FA and glucose metabolism, cellular proliferation, and differentiation, as well as inflammation [236,237]. In this regard, a variety of PUFAs and their oxidized metabolites (epoxy- or hydroxy-PUFA) are potential ligands that can bind to PPARs [238]. Studies have demonstrated that PUFAs bind to PPAR isoforms in the μM range, while oxidized metabolites of PUFAs can activate PPARs at a nM level [239,240]. These data suggest that the oxidized metabolites of PUFAs could be endogenous PPAR agonists.

An in vitro study showed that lipopolysaccharide (LPS)-induced inflammation in astrocytes downregulates CYP2J3 and CYP2C11, which are the major CYPs to produce epoxy-PUFAs. Interestingly, inhibition of NF-κB restores expression of CYP2J3 and has a limited effect on CYP2C11, suggesting that additional regulatory mechanisms are involved in CYP2C11 regulation during LPS-induced inflammation [241]. One of the alternative mechanisms might be PPAR α pathways. PPARα is one of the three ligand-activated transcription factors of PPARs, a subfamily of the nuclear receptor superfamily [242].

Several studies demonstrated that PPAR-α agonists down-regulate CYP2C11. which is a major CYP involved in the production of Ep-PUFAs. This result suggested that a PPAR-α agonist could affect neuroinflammation through modulating the endogenous level of Ep-PUFAs [243,244,245]. On the other hand, Wary et al. demonstrated that overexpression of human CYP2J2 in HEK293 cells has inductive effects on all three subfamilies of PPAR (i.e., PPARα, -β/δ, and -γ) reporter gene activity [106]. Furthermore, IL-1β-induced NF-κB reporter activity was significantly inhibited in cells expressing the specific combination of CYP2J2 and PPARα. They also found that the 8,9-EET, and 11,12-EET, but not 14,15-EET could activate PPARα [106]. Consistent with this, Node et al. found that 11,12-EET has anti-inflammatory effects, and suggested PPARα as an anti-inflammatory target for 11,12-EET and CYP2J2 [105]. Liu et al. also found that treating endothelial cells with EETs or AUDA leads to an anti-inflammatory response and further treatment by GW9662, a PPARγ inhibitor, significantly abolished the EET-mediated anti-inflammatory effect, potentially by preventing IκB degradation [246]. Other Ep-PUFAs have also been shown to activate PPAR. For instance, PPAR activation may also be possible, as 17,18-EEQ inhibits TNFα -induced inflammation in human bronchi, which is sensitive to PPARγ inhibition [247]. These studies suggest that PPARs could be potential receptor targets for Ep-PUFAs in regulating the inflammatory response. However, how PPAR activation could mediate the effects of EETs is still unclear, mainly because some of the early sEH inhibitors could also cause PPARα activation through off-target effects [248].

#### 7.3.2. TRP Channels

Transient receptor potential channels are non-selective cationic channels, which can control various cellular functions [249]. These channels, upon activation, depolarize the cell, which leads to activation or inactivation of specific voltage-gated ion channels and Ca^2+^ homeostasis. TRP channels, especially TRP vanilloid (TRPV), TRP ankyrin (TRPA), and TRP canonical (TRPC) are highly expressed in the brain and play significant roles in brain development, synaptic transmission, neurogenesis, and neuroinflammation by regulating neuronal and glial functions [250,251,252,253]. The molecular mechanism of TRP channel regulation is yet to be elucidated. However, several findings suggested that their activity is modulated by various PUFA metabolites, in particular Ep-PUFAs [249,254].

Several studies indicated different Ep-PUFAs as potential ligands for TRP channels. Watanabe et al. found that 5,6-EET activates TRPV4 in murine endothelial cells to induce hyperpolarization [255]. Vriens et al. also implicated 5,6-EET and 8,9-EET as direct TRPV4 agonists by studying mouse aortic endothelial cells. They also showed that pharmaceutical modulation of CYP2C9 induced robust Ca^2+^ responses by TRPV4 stimuli such as AA and cell swelling, while CYP2C9 inhibition abolished the response to stimuli. None of these effects were observed in TRPV4^−/−^ mice [256]. In addition, some studies have shown that 11,12-EET can stimulate TRPV4 in cerebral artery smooth muscle cells leading to hyperpolarization, an increase in the BKCa on the glial cell membrane, and regulation of neurovascular coupling [257,258]. Furthermore, using in vivo screening in a transgenic *C. elegans* model expressing rat TRPV4, Caires et al. indicated that 17,18-EEQ is necessary for the function of this channel [259]. Furthermore, genetic ablation and diet supplementation revealed a decrease in TRPV4 activity when PUFAs and related eicosanoid levels were reduced [259].

Interestingly, studies found that EETs can bind to multiple TRP channels independent of Ca^2+^ signaling [260,261]. For instance, even though 5,6 EET can activate TRPV4 in colonic afferents and potentially cause visceral hyperalgesia, it acts as a TRPA1 activator and causes a TRPV4-independent Ca^2+^ transient in the lumbar DRG neurons L4 and L5 [262,263,264]. Sisignano et al. also observed an increase in 5,6-EET levels of the dorsal spinal cord and dorsal root ganglia (DRG) during capsaicin-induced nociception [264]. They also found that 5,6-EET treatment can induce calcium flux in cultured DRG neurons, while this treatment has no effect in TRPA1 negative cells [264]. Brenneis et al. found that 8,9-EET can significantly increase the amplitude of allyl isothiocyanate- (AITC) induced calcium increases in cultured DRG neurons through activation and sensitization of TRPA1 channels [265]. Liu et al. demonstrated that sEH inhibition of aortic vascular smooth muscle cells with 8-HUDE could increase vascular tone, calcium flux, and upregulation of TRPC1 and C6 [266]. Interestingly, another study by Fleming et al. showed that 11,12-EET enhances intracellular translocation of activated TPRC6 channels to the plasma membrane, indicating a possible connection to hyperpolarization [267]. 11,12 EET was also reported to activate a TRPV4-TRPC1-BKCa complex in smooth muscle cells [268], but it remains to be shown whether such channel complexes also exist in neurons. These studies reveal the complexity of how Ep-PUFAs affect TRP channels with multiple possible mechanisms, which depend on the compound, tissue tested, and receptor expression.

#### 7.3.3. GPCRs

G-protein coupled receptors (GPCRs) such as chemokine receptors, eicosanoid receptors, histamine receptors, and adenosine receptors are the largest family of membrane-bound proteins, and of more than 370 identified non-sensory GPCRs, about 90% are expressed in the brain [269,270]. The role of GPCRs in the brain includes, but is not limited to, appetite, mood, inflammation, pain, synaptic transmission, as well as cognition [271]. Furthermore, these receptors are involved in the pathogenesis of several neurodegenerative diseases such as AD, PD, and HD [269,270,272]. A more comprehensive discussion on the GPCRs involved in the pathogenesis neuroinflammation and neurodegeneration is covered in related reviews [272,273,274].

Even after decades of research, the GPCRs for Ep-PUFAs have yet to be identified. In this regard, Wong et al. identified a specific high affinity binding site for 14,15-EET through a series of studies on guinea pig mononuclear (GPM) and human monocyte (U937) cells [275,276,277]. These studies revealed that the potential binding site of 14,15-EET is saturable and regio- and stereoselective, with the highest affinity for 14(R), 15(S)-EET compared to other enantiomers. The binding is also sensitive to protease treatment, which indicates that the target is a protein, and is associated with a receptor that can be downregulated by an increase in intracellular cAMP and activation of a protein kinase A signal transduction mechanism, which is consistent with the mechanism of action of GPCRs for other prostanoids [275,276,277]. Interestingly, elevated levels of cAMP, associated with the PKA activation, are able to inhibit NF-kB, exerting anti-inflammatory activity. Therefore, this pathway may explain the anti-inflammatory activity of EETs [278]. In addition, Chen et al. also demonstrated that EETs, in particular 14,15-EETs, can stimulate membrane-associated activities through activation of Src and initiation of a tyrosine kinase phosphorylation cascade [279]. Snyder et al. also demonstrated that a 14, 15 EET analog coupled with silica beads, which can only act on the cell surface, completely retained its ability to inhibit cAMP-stimulated aromatase activity. These data further support possible GPCR targets for 14,15 EETs [280]. Behm et al. suggest that EETs may function as endogenous GPCR antagonists, which can cause anti-inflammatory actions by direct inhibition of thromboxane receptors [281]. However, even after decades of research, researchers have yet to find a specific high affinity GPCR for 14, 15 EETs using an array of isolated tissue/cell biochemical assays and GPCRs screening [281,282].

Besides 14,15-EETs, the other regioisomers of EETs, 11,12-EET, has also been extensively studied with the hope of identifying a GPCR [108,283,284,285]. It should be mentioned that in all these studies, non-neural cells were used as target cells, implying that this observation might not be applicable to neuronal cells, as the cell type used can be a key factor in determining the type of GPCR activation by EETs. Keeping these ideas in mind, Mule et al. used CA1 pyramidal cells (PCs) in the mouse hippocampus and demonstrated that the 11,12 EET mediated opening of a G protein-coupled inwardly-rectifying potassium (GIRK) channel through the activation of Gi/o proteins, introducing a new EETs-dependent cellular pathway, which remains to be elucidated [286]. The activation of Gi/o proteins corroborate other studies suggesting that EETs are a potential target for the μ-opioid receptor, a GPCR system mainly linked to Gi/o proteins [287,288].

#### 7.3.4. ER

ER stress occurs when the homeostatic protein folding and trafficking in the cell are overwhelmed or unbalanced, leading to the unfolded protein response (UPR) and often to apoptosis [177]. The failure of ER adaptive capacity, which is a hallmark of several neuroinflammatory disorders such as PD and AD, can be induced by different factors including, but not limited to, neurotoxins, unfolded and misfolded proteins, mitochondrial ROS, and high glucose as in diabetes [289,290,291]. ER stress intersects with many different inflammatory and stress signaling pathways, such as the NFκB pathway, and thus the therapeutic mechanism(s) of action of Ep-PUFAs may be related to their interactions with the molecular constituents therein [292]. Several studies have shown the effects of Ep-PUFAs on the phosphorylation state of the key ER stress proteins including inositol-requiring enzyme (IRE1α), protein kinase R-like endoplasmic reticulum kinase (PERK), and activating transcription factor 6 (ATF6) (Figure 6B). For instance, Bettaiab et al. observed that a high fat diet (HFD)- and chemical-induced ER stress increased sEH expression. Meanwhile, sEH ablation or its pharmacological inhibition attenuated this stress by decreasing the phosphorylation of IRE1α, PERK, and ATF6 [293]. In addition, Ren et al. found that administration of TPPU or sEH deletion suppressed these markers in the striatum of MPTP-treated mice [19].

Even though the actual mechanism behind the Ep-PUFAs effects on the ER stress pathway remains to be revealed, there are several points of evidence. EETs may act partly through the PI3K/ Akt pathway. For instance, Dhanasekaran et al. demonstrated that treatment of HL-1 cardiac muscle cells and HL-1 cardiac muscle cells with 14,15-EETs under hypoxia/reoxygenation conditions increased AKT phosphorylation PIP3 level, indicating that EETs can act at least partly through the PI3K/AKT pathway [294]. ER-stress suppression through AKT activation prevents apoptosis by upregulating apoptotic protein inhibitor family proteins such as XIAP, cIAP-2, and mitochondrial Bcl-2 [295,296,297]. Ep-PUFAs can also act through JNK and p38 MAPK pathways. Inceoglu et al. observed that kinase mediators of neuropathic pain, p38, and JNK can be effectively blocked by inhibition of sEH [298]. Despite this evidence, the direct interaction between Ep-PUFAs and the ER has not yet been reported. Thus, a better understanding of the nature of the involvement of Ep-PUFAs on ER stress can lead to new therapeutic strategies to reduce the incidence of neuroinflammatory disorders.

#### 7.3.5. Mitochondria

In an effort to find the underlying mechanisms of age-associated neurodegeneration, the main hypothesis is impairment in protein quality control systems and protein accumulation [1,221]. An alternative mechanism that recently emerged is the loss of mitochondrial function and uncontrolled production of ROS leading to neurodegeneration, particularly in PD and AD development. For instance, it has been shown that age-related mitochondrial dysfunction affects both the expression and processing of amyloid-β protein precursor (AβPP), which initiates the Aβ accumulation, the primary hallmark of AD [299]. Furthermore, there is evidence of disruption in mitophagy (a selective and programmed degradation of damaged mitochondria by autophagy) by aging, leading to an increase in the damaged mitochondrion, which can also contribute to age-related neurodegeneration [299,300]. Mitochondrial dysfunction has been well documented in AD and PD [1,2,299,300,301].

Recent studies have revealed the promising potential of Ep-PUFAs in rescuing mitochondrial function in various models [302,303]. For instance, Samokhvalov et al. indicate that 14,15-EET can protect mitochondrial functions in starvation-induced injury, probably through regulation of the autophagic response [304]. Katragadda et al. also found that exogenous supplementation of 14,15-EET protects mitochondrial function and reduced cellular stress in isolated rat cardiomyocytes and H9c2 cells, which is suggested to be through the mitoK^+^ channel-dependent pathway that preserves mitochondrial membrane potential under cellular stress [305]. Wang et al. also demonstrated that 14,15-EET could protect cortical neurons from apoptosis in oxygen–glucose deprivation (OGD) condition by elevating mitochondrial biogenesis, and acts through activation of PGC-1α and NRF-1 mediated by cAMP response element-binding protein (CREB) [303].

EETs have also been shown to preserve mitochondrial function through mitigating the uncontrolled release of ROS and decreasing ER response (Figure 6B). All of these attenuate cellular damage that is accompanied by neuroinflammatory pathology [302,306,307]. EETs have been shown to induce some cellular mechanisms to buffer free radicals, hence protecting sub-cellular organelles from oxidative damage [193,308]. Liu et al. indicated that 11,12-EET-pretreatment of carcinoma cells could attenuate the ROS-mediated mitochondrial dysfunction through induction of antioxidant proteins superoxide dismutase and catalase, thereby mitigating different major apoptotic pathways such as activation of p38 MAP kinase, c-Jun NH2-terminal kinase, as well as caspase-3 and -9 [309]. Qu et al. also found an anti-apoptotic effect of 14, 15-EET in cerebral microvascular smooth muscle cells under OGD conditions by reducing the free radical level through the JNK/c-Jun and mTOR signaling pathways [310]. Nevertheless, the specific signaling pathways that result in direct effects of EETs on mitochondrial function and ROS levels in glia and neuronal cells are still unknown.

## 8. Future Directions

PUFAs are an emerging class of dietary components that play a critical role in aging and neurodegeneration. However, the effect and the underlying mechanism of PUFAs in neurodegeneration remain largely unknown. In this review, we discussed that downstream CYP PUFA metabolites, whose endogenous levels are greatly affected by diet, are largely beneficial for neurodegenerative diseases. Studies have shown that the anti-inflammatory effects of CYP PUFA metabolites, particularly Ep-PUFAs, could alleviate neuroinflammation, which is beneficial in neurodegenerative diseases. Ep-PUFAs enhance the production of anti-inflammatory cytokines and decrease the production of pro-inflammatory cytokines in murine models of Alzheimer’s disease and Parkinson’s diseases. While the beneficial effects of Ep-PUFAs in NDs have been well established with inhibition of sEH, which hydrolyzes endogenous Ep-PUFAs, the molecular mechanisms have not been solved. As discussed above, there are several proposed molecular mechanisms through which Ep-PUFAs modulate neurodegeneration. Understanding the molecular signaling mechanism of Ep-PUFAs in neurodegeneration could help us to design a better diet for the aging population and could identify potential new therapeutic targets for neurodegenerative diseases.

Studying the molecular mechanisms of neurodegeneration is difficult due to their complexity and lack of a translatable in vitro model. As discussed, it becomes even more challenging with Ep-PUFAs because their protein target has yet to be identified. To date, a better way to study neurodegeneration remains animal models; however, these models are expensive and low throughput. Fortunately, due to the recent development of novel genetic models, including *C. elegans* and zebrafish, our understanding of the mechanism of neurodegeneration can be greatly accelerated. Both *C. elegans* and zebrafish are highly homologous to humans, and most of the disease-causing genes in humans have the orthologues in both *C. elegans* and zebrafish. These models will facilitate the investigation of the molecular mechanism(s) driving neurodegeneration. Our laboratory is currently studying the mechanism of CYP PUFA metabolism in neurodegeneration using *C. elegans* as a biological model.

As we discussed, there is an unmet need to develop novel therapies for neurodegenerative diseases. In this review, we have shown that sEH could be a novel target for Alzheimer’s disease and Parkinson’s disease. However, there are challenges in developing sEH inhibitors to treat NDs. Although the sEH inhibitors used in the murine model can cross the BBB, they require further optimization to better cross the BBB with improved physical properties for formulation. Ep-PUFAs are a class of CYP PUFA metabolites that contains both ω-3 and ω-6 Ep-PUFAs. However, in the reported studies, the active Ep-PUFAs have yet to be identified. Since the endogenous levels of ω-3 and ω-6 Ep-PUFA are greatly impacted by the diet, the beneficial effects of an sEH inhibitor for NDs may be greatly impacted by patient diets. Therefore, an alternate approach for developing a new therapy for neurodegenerative diseases would be designing a mimic of the active Ep-PUFAs that would not be affected by diet or ω-3 supplementation. All in all, CYP PUFA metabolism is an exciting pathway for research in neurodegeneration, and more research is needed to better understand the mechanism(s) behind the effects of CYP PUFA metabolites in neurodegeneration.

## Figures and Tables

**Figure 1 nutrients-12-03523-f001:**
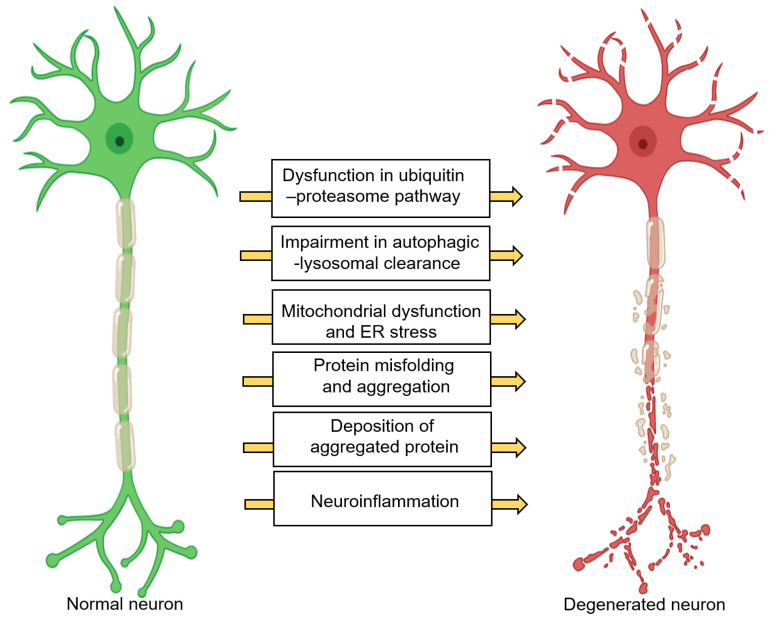
Common neuronal pathways that are changed in different neurodegenerative diseases (1) Dysfunction in the ubiquitin–proteasome pathway increases the intracellular misfolded, damaged, or unneeded protein. (2) Dysfunction in the autophagy–lysosomal pathway triggers the accumulation of pathogenic protein aggregates and damaged mitochondria. (3) Mitochondria dysfunction causes dysfunction and uncontrolled release of reactive oxygen species. (4) ER stress occurs when the homeostatic protein folding and trafficking in the cell is overwhelmed or unbalanced, leading to UPR. (5) Transcellular propagation and seeding of protein aggregates cause disease progression. (6) The aggregation of misfolded proteins contributes to toxicity. (7) A chronic neuroinflammatory state can lead to protein accumulation, ER stress, mitochondria dysfunction, uncontrolled oxidative stress, and axonal transport impairment. Abbreviations- ER: endoplasmic reticulum, UPR: unfolded protein response.

**Figure 4 nutrients-12-03523-f004:**
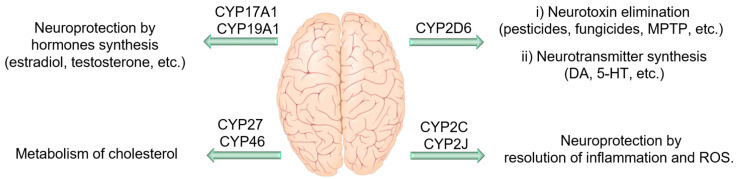
The protective role of some cytochrome P450s (CYPs) in neurodegenerative diseases. Abbreviations- CYP: cytochrome P450, DA:dopamine, 5-HT: 5-hydroxytryptamine (or serotonin), MPTP: 1-methyl-4-phenyl-1,2,3,6-tetrahydropyridine, ROS: reactive oxygen species.

**Figure 5 nutrients-12-03523-f005:**
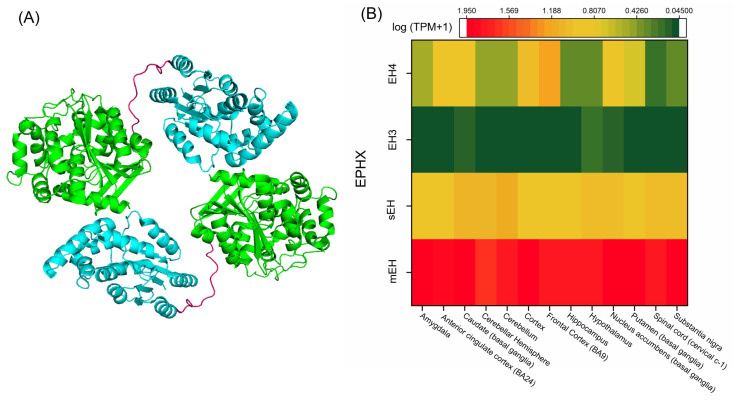
(**A**) Crystal structure of the sEH dimer (PDB accession code 1S8O) [150]. The sEH monomer is composed of two globular regions representing the α/β tertiary structure, and a short proline-rich linker (pink) connects the C-terminal of one monomer to the N-terminal region of another. The catalytic site with epoxide hydrolase activity is located within the C-terminal region, whereas the N-terminal region possesses phosphatase activity. (**B**) EH distribution in the human brain based on mRNA expression. Expression values are shown as a median of TPM, calculated from a gene model with isoforms collapsed to a single gene with no other normalization [152]. Abbreviations- EH: epoxide hydrolase, EH3: epoxy hydrolase 3, EH4: epoxy hydrolase 4, *EPHX*: epoxide hydrolase gene, mEH: microsomal epoxide hydrolase, PDB: Protein Data Bank, sEH: soluble epoxide hydrolase, TPM: transcripts per million.

**Figure 6 nutrients-12-03523-f006:**
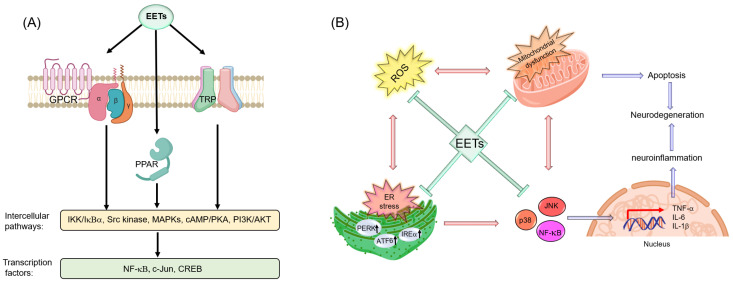
(**A**) Mechanisms of action of EETs through three main cellular targets. Verity of EET functions occur through PPAR-, TRP-, and GPCR-dependent mechanism, which activates intercellular pathways that modulate transcription factors in the target cell; (**B**) The possible anti-inflammatory mechanism of EETs. Abbreviations- ATF6: activating transcription factor 6, EETs: epoxyeicosatrienoic acids, ER: endoplasmic reticulum, GPCR: G protein-coupled receptor, IRE1α: inositol-requiring enzyme alpha, PERK: protein kinase R-like endoplasmic reticulum kinase, PPAR: peroxisome proliferator-activated receptor, TRP: transient receptor potential channel, ROS: reactive oxygen species. Figure is in some parts created with biorender.com.

**Table 1 nutrients-12-03523-t001:** Different residues in the catalytic triad of mEH, sEH, EH3, and EH4 and their function.

EH Type	Polarizing Residues	Nucleophilic Residue	Involved in Orientation and Activation of Nucleophilic Residues
mEH	Tyr299 Tyr374	Asp226	His431 Glu404
sEH	Tyr383 Tyr466	Asp335	His524 Asp496
EH3	Tyr220 Tyr281	Asp173	His337 Asp307
Eh4	Tyr216 Tyr281	Asp169	His336 Asp307

Abbreviations—Asp: Aspartic acid, EH: epoxide hydrolase, EH3: epoxy hydrolase 3, EH4: epoxy hydrolase 4, Glu: Glutamic acid, His: histidine, mEH: microsomal epoxy hydrolase, sEH: soluble epoxy hydrolase, Tyr: Tyrosine.

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
