# Peer review of "Cytochrome P450 Metabolism of Polyunsaturated Fatty Acids and Neurodegeneration"

_nutrients, 2020, doi:10.3390/nu12113523_

Round 1

Reviewer 1 Report

The Authors have written a very good review regards Cytochrome P450 .

I can see from my perspective that authors have written a very comprehensive review regards the metabolism of polyunsaturated fatty acid and it’s effect on neurodegeneration . I think this review is novel , as the author discuss in this review about the effect of cytochrome metabolism on neurodegeneration. This review will be beneficial for other authors working on Cytochrome inhibition and the metabolism of polyunsaturated fatty acid in the body

Author Response

We would like to thank the referees for their constructive comments and suggestions on the manuscript entitle “Cytochrome P450 metabolism of polyunsaturated fatty acids and neurodegeneration” (Manuscript ID: nutrients-969179). We have made careful revisions based on the feedback, and a detailed response to the comments is listed below.     

Reviewer 1

The Authors have written a very good review regards Cytochrome P450. I can see from my perspective that authors have written a very comprehensive review regards the metabolism of polyunsaturated fatty acid and it’s effect on neurodegeneration . I think this review is novel, as the author discuss in this review about the effect of cytochrome metabolism on neurodegeneration. This review will be beneficial for other authors working on Cytochrome inhibition and the metabolism of polyunsaturated fatty acid in the body.

Response: We would like to thank the reviewer 1 for carefully reviewing the manuscript and for  their positive feedback.

Reviewer 2 Report

The manuscript by Sarparast et al., discusses the role of polyunsaturated fatty acids and neurodegenerative diseases including Alzheimer’s disease and Parkinson’s disease. This is overall a  very interesting paper and links several fields of the epoxygenase pathway. The authors thoroughly discuss how Cytochrome P450 epoxygenases can form epoxide metabolites, and how these are implicated in neurodegenerative diseases. The authors were also very comprehensive in their discussion by introducing aspects of formation as well as degradation of PUFA metabolites, and their roles in ND. However, the manuscript in some sections are congested with information without a clear focus. Hence, the authors should add lines that sums up the major finding of the paper. For instance, a dedicated section discussing enantiomers would be greatly beneficial. Additionally, the major CYP’s responsible for PUFA metabolism should focus primarily on CYP’s, rather than the metabolites and possible stereoisomer outcomes. Lastly, the manuscript discusses topics focused around age-related neurodegenerative diseases including PD and AD. While some of the topics discussed could apply to other neurodegenerative diseases, please be mindful to not overstate the implications of these PUFA metabolites and other neurodegenerative diseases. Nevertheless, the manuscript reads very well and is very thorough in their discussion on how CYP450 metabolism of PUFA’s may be implicated in neurodegenerative diseases. It will be very beneficial if the authors could write a section on how the knowledge from this paper can help in future research design in this research area. The references does not include several new findings on this area of work. 

  • Overall comment: one section clearly describing the known pathogenesis and pathways of neurodegenerative diseases would be greatly beneficial to prime the reader.
  • Line 21: Please change ‘in generally” to “in general”
  • Line 31: It is true that aging is the primary risk factor for most neurodegenerative diseases. However, the definitive claim that neurodegenerative diseases are age-related is an overstatement. Please be mindful of this. In the rest of the first introductory paragraph, the authors state that no curative treatment has been developed because of the lack of understanding of the mechanisms. It appears that this paragraph was written in such a way to lead into age-associated neurodegenerative diseases. Line 31 however suggests that all ND’s are age-related.
  • Line 109: please change “w” in “w-3” into the ω This remains true for the rest of the manuscript.
  • Line 148: PUFAs as written in the introduction include ω-6 and ω-3 fatty acids. Thus the overarching classification that PUFAs are metabolized by the arachidonic acid cascade (suggesting specificity for ω-6 fatty acids), may not be appropriate in this case.
  • Line 200: Please capitalize ‘M’ in the title word, ‘Major’
  • Line 200: The section 3.3 Major CYP responsible for PUFA metabolism do not seem to focus much on the CYP’s that metabolize PUFA’s, but rather the effects of different stereo- and regio- isomers. The discussion regarding regio- and stereo- isomers may be best suited in a section of its own. The other information remains important, and should be highlighted more in this section for the different roles of CYP’s.
  • Line 533: While not incorrect, it is important to recognize that (chronic) activation of glial cells in response to a stimuli or antigen leads to the chronic neuroinflammatory state. It can be considered less a hallmark, and more the source of neuroinflammatory responses.
  • Line 716: A dedicated section regarding the enantiomers seem appropriate. There remains a lot to be discussed, and a focused section highlighting what is known and not known may help bring a concentrated attention to the topic of stereochemistry.
  • Line 838: Please add in “R” at the end of the word ‘inhibitor’

Author Response

We want to thank the referees for their constructive comments and suggestions on the manuscript entitled “Cytochrome P450 metabolism of polyunsaturated fatty acids and neurodegeneration” (Manuscript ID: nutrients-969179). We have made careful revisions based on the feedback, and detailed responses to the comments are attached.

Reviewer 3 Report

The review reports a comprehensive and detailed description of the role of polyunsaturated fatty acids (PUFAs) in neurodegenerative disease and in neuroprotection, providing interesting clue on possible mechanisms of action of the cytochrome P450 epoxidized-PUFA products which are beneficial for the brain.

Minor issues:

  • Lines 35-37, page 1: “Despite decades of effort, no curative treatment has been developed for these diseases, and almost all medication interventions are aimed at reducing the symptoms of these diseases”, this latter “of these diseases” is redundant and can be removed.
  • Lines 39-40, page 1: “Human genetic studies revealed several genes responsible for NDs, such as APOE, which…”, this is the first time this acronym appears, please write the full name.
  • Omega-3 PUFA and omega-6 PUFA are sometimes written as full name, sometimes as greek letter, sometimes as a “w”. Please, write the full name only the first time they are mentioned then always keep the greek letter nomenclature.
  • Line 58, page 2: “DAergic neurons”, even if it is clear that authors are referring to dopaminergic neurons it is better to write it full.
  • Line 77-79, page 2: “The beneficial properties of Ep-PUFAs are lost when Ep-PUFAs are converted to their corresponding diols by soluble epoxy hydrolase (sEH) [17].” The reason for this event to happen will be largely explained and described later in the review. Here it only misses the clarification that this mechanism will be largely discussed later adding a final comment such as “as it will be better discussed later”.
  • Lines 158-164, page 4: “CYPs in general contain (i) signature residue sequence of 158 FXXGXbXXCXG, in which Xb is a basic residue and the cysteine is located at the axial position to the heme; and (ii) a soret peak at 450 nm when a carbon monoxide binds to the Fe(II) in of the heme group [41,42]. Note that there are some other proteins with the same heme group, axial cysteine residue, similar sSoret peak in presence of CO, as well as some related catalytic properties such as some peroxidases and nitric oxide synthases, but they are not considered as CYP. Some proteins. The 3D structures of these proteins also differentiate them of CYP enzymes [43,44]”. Mistakes and corrections are in red bold and underlined.

Here authors should underline the fact that CYP enzymes all share the same fold.

  • Lines 174-177, page 4: “Of the 57 human CYP enzymes, 50 are located on ER and are usually involved in xenobiotics metabolism (i.e., drugs and environmental pollutants), while the rest are is located in the mitochondria membrane and is generally engaged in metabolism/biosynthesis of endogenous molecules 176 [49]”. Mistakes and corrections are in bold and underlined.
  • Lines 181-183, page 4: “Historically, the first CYP enzyme was described by Klingenberg and Garfinkel as an unknown pigment that binds carbon monoxide in its reduced form and produces a soret absorption peak at 450 nm [51,52]”. Mistakes and corrections are in bold and underlined.
  • Line 187, page 4: “CYPs catalyze a large variety of reactions including oxidation of heteroatom, heteroatom dealkylation, C–C bond…”. Mistakes and corrections are in bold and underlined.
  • Lines 190-199, pages 4-5: “Figure 2 shows the monooxygenase mechanism (here author should specify that the mechanism reported in Fig2 is specific for an epoxidation reaction) of CYPs which can be divided in 7 steps: 1) Before binding the substrate to CYP protein, there is an equilibrium between the hexa- and pentacoordinate Fe(III); substrate binding to the CYP enzymes shifts the equilibrium in favor of pentacoordinate.; 2) an electron will transfer to this complex either directly from NAD(P)H or through a redox protein partner, to reduce Fe (III) to Fe (II). Note that this step is critical for substrate oxidation, as the diatom oxygen cannot bind to Fe (III); 3) oxygen binds to the Fe(II); 4) the next electron is either transferred directly from NAD(P)H or through a redox protein partner; 5) two subsequent protonations occur; 6) the complex gets deprotonated by releasing a water molecule, which results in an iron-oxo complex; 7) Ffinally, the oxygen group atom is transferred to the substrate and results in an oxidized products [57–59].
  • Line 200, page 5: “3. major CYP responsible for PUFA metabolism”, use the capital letter to start the sentence of the title.
  • Line 205-206, page 5: “….involved in oxygen insertion (Figure 1B, 5,6-EET, 8,9-EET, 11,12-EET, and 14,15-EET; EET); each of the EETs products can be either the R,S- or the S,R- stereoisomer [61,62]”.
  • Line 217, page 5: “yields 19-HETE as the predominant product, and 16-, 17- and 18-217 HETE as minor product [22,60]”. Something is missing in this sentence.
  • Line 238, page 5: “…and involved in biosynthesis of serotonin and dopamine, are mainly 238 expressed in the brain [73,74];…”; something is missing at the beginning of the sentence, maybe “CYP2D6 that is…”.
  • Line 248, page 6: “Mann et al., has shown….
  • Line 262-263, page 6: “Thus, epoxide could covalently react with nucleophile and caused a wide range of biological and pathological effects.”
  • Line 286, page 6: “Among the EPHXs EHs, mEH is the first…”.
  • Line 287-288, page 6: “This membrane-bound enzyme is attached to the surface of the ER or the plasma membrane by its N-terminal membrane anchor [95,96].
  • Lines 307-310, page 7: “EH3 and EH4 have 45% homology in their sequence and were originally named as α/β hydrolase domain containing protein 9 (ABHD9) and protein 7 (ABHD7), but were renamed after studies done by Arand’s group showed that epoxide hydrolase activity toward epoxy octadecenoic acids (EpOMEs) and EETs [103,104]”.
  • Lines 314-315, page 7: “while the highest expression was observed in skin, stomach, lung, and tongue [104]”.
  • Lines 326-327, page 7: “EH4 expression is much higher in the brain compared to other tissues..”.
  • Lines 332-335, page 7: “However, the presence of a nucleophilic aspartate in the catalytic triad (that distinguishes EH3 and 4 from other ABHDs), and high homology with two EHs in C. elegans [17], suggests the potential for EH4 to be an active 334 EH with important physiological activity in the brain [103]”.
  • Lines 336-341, page 7: “4.2. Catalytic function and mechanism of EHs:

The major function of EH is to hydrolase hydroxylate xenobiotic epoxides and EpPUFAs. (Figure 1B). The catalytic activity of EHs can be described in three steps, with the contribution of four amino acids, which are involved in transforming the epoxide to a diol [85]. In the first step, the epoxide enters through the L-shaped hydrophobic tunnel with the nucleophilic aspartate in the center and it is stabilized there by Vvan der Walls and hydrogen bonding with two tyrosines and the hydrophobic pocket with the hydrophobic pocket and two tyrosines, respectively”.

  • Line 351, page 8. Authors refer to a Table (named Table x) that does not appear in the manuscript. Be sure to insert it in the final version and number it.
  • Line 352-353, page 8: “Besides these highly conserved regions, the amino acid sequences and the N-terminal part are largely different a…”.
  • Lines 356-358, page 8: “sEH has been the main enzyme that degrades Ep-PUFAs but other EHs also play a role in Ep-PUFA metabolism. [85]. mEH can selectively hydrolase arene and cyclic epoxides, is involved in 357 xenobiotic metabolism, and catalyzes the epoxidation on a wide and still growing range of substrates that is still growing [102]”.
  • Line 361, page 8: “(kcat./Km)”.
  • Line 379-382, page 8: “The role of mEH in neurological disorders is a subject of intense interest in the field of pathophysiology-related EPHX1 due to its high expression in the brain [102]. In supporting of this, Lui et al., identified high expression of mEH in the hippocampus and in the cortex of AD patients [115]”.
  • Line 386, page 8: “…further supporting the role of mEH role in the pathogenesis of neurodegeneration…”.
  • Line 387-388, page 8: “Generic ablation of mEH in mice treated with methamphetamine (METH) caused an decreased in synaptosomal DA uptake in the striatum, …..”.
  • Line 394, page 8: “Even though mEH possesses low catalytic activity….”.
  • Line 395-396, page 9: “high levels of mEH expression in specific brain regions and neurons, as well as evidence for substrate channeling between mEH and CYP on ER, suggests that mEH contributes to….”.
  • Line 402, page 9: “In short, while mEH activity in hydrolyzing Ep-PUFAs is well studied, their its role in neuronal….”.
  • Line 406-407, page 9: “…oligodendrocytes, and neural cell bodies. [123,124]. sEH expression in microglia cells has not yet been well characterized, however, Huang et al., found sEH expressed in the BV2….”.
  • Line 411, page 9: “and neurodegenerative pathogenesis involving…”.
  • Line 420, page 9: “Hoopes et al., used EH3 knockout….”.
  • Lines 431-434, page 9: “Brain has high capacity for de novo synthesis of the epoxyEp-PUFAs from their parent PUFAs by CYP, which can act on neuronal cells, and the Epoxy-PUFAs are rapidly degraded by EHs such as sEH[131,132] . While the CYP pathway seems to be the dominant route of production of Ep-PUFAs in brain, they can be also presence in transported to the brain by cellular uptake [133,134].”
  • Lines 449-452, page 10: “Ostermann et al. found that an ω‐3 PUFA enriched with EPA and DHA reduced Ep‐449 PUFAs and the ratio of epoxy‐ to dihydroxy‐PUFA, revealing the higher sEH activity [139]. Rey at al. also observed that a dietary ω-3 PUFAs supplementation promotes synthesis of proresolving oxylipins in the mice hippocampus [140]”.
  • Line 458, page 10: “derived oxylipins in CNS system.”.
  • Line 496, page 11: “…4-aminopyridine as a potassium channel blocker. ; Vito et al. reported…”.
  • Line 512, page 11: “AUDA (an sEH inhibitor),….”.
  • Line 518, page 11: “signaling;, and the effects were abolished”.
  • Line 527, page 11: “… and by direct neuro-protective effects on the neurons through…”.
  • Line 550, page 12: “….a region- and cell-specific manner, which was suggested to be due….”.
  • Line 605, page 13: “…EET, could attenuate the neurotoxicity induced by MPTP [177]. Ren et al., also demonstrated that…”.
  • Lines 610-611, page 13: “In addition to these studies, several studies has data have shown that the sEH 610 expression positively correlates with phosphorylation of α-synuclein in the striatum, which…
  • Line 618, page 13: “….shown that EETs initiate its their anti-inflammatory effects…”.
  • Line 653-655, page 14: “Liu et al. also found that treating endothelial cells either by EETs or AUDA increases the anti-inflammatory response, and further treatment by GW9662 a PPARγ inhibitor) significantly abolished the EET-mediated anti-inflammatory effect, potentially….”.
  • Line 665, page 14: “These channels, upon activation, depolarize the cell…”.
  • Lines 682-685, page 14: “…..screening in a transgenic C. elegans model expressing rat TRPV4, Caires et al. indicated that that 17, 18 epoxyeicosatetraenoic acid (17,18-EEQ) are necessary for the function of this channel[204]. Also, 683 genetic ablation and diet supplementation revealed a decrease in TRPV4 activity when PUFA and 684 related eicosanoid levels were reduced [204]”.
  • Lines 686-689, page 14: “Interestingly, studies found that EETs can bind to multiple TRP channels independent of Ca2+2+ signaling [205,206]. For instance, even though 5,6 EET can activate TRPV4 in colonic afferents and potentially cause visceral hyperalgesia, it acts as a TRPA1 activator and causes a TRPV4-independent Ca2+ transients (did the authors mean “transfer”?) in to the lumbar DRG neurons L4 and L5 [207–209]”.
  • Lines 700-701, page 15: “…but it remains to be shown whether such channel complexes are also exist in neurons.
  • Line 709, page 15: “synaptic transmission, as well as cognition [216];. Also, these receptors…”.
  • Line 763, page 16: “…increased AKT phosphorylation PIP3 level, indicating that EETs can act at least partly act through…”.
  • Lines 814-819, page 17: “While the beneficial effects of Ep-PUFAs in neurodegenerative diseases have been well established with the inhibition of she sEH, which hydrolyzes endogenous Ep-PUFAs, the molecular mechanisms have not been solved. As we discussed above, there are several proposed molecular mechanisms behind how Ep-PUFAs modulate neurodegeneration. Understanding the signaling mechanism of Ep-PUFAs in neurodegeneration could help us in designing a better diet for the aging population and could identify potential new therapeutic targets for neurodegenerative diseases”.
  • Lines 833-836, page 17: “However, there are challenges in developing sEH inhibitors to treat neurodegenerative diseases.: 1) Although the sEH inhibitors used in the murine model can cross blood brain barrier (BBB), they requires further optimization to be more BBB permeable with improved physical properties for formulation.; 2) Ep-PUFA is a class…..”.
  • Line 838, page 17: remove the comment.
  • Line 839, page 17: “…the beneficial effects of an sEH inhibito inhibition for neurodegenerative diseases may….”.
  • As a general comment, authors should take better care of the usage of abbreviations. They should write the full name only when something is mentioned the first time, then they should always keep the abbreviated name. There is no reason to rewrite the full name of something, and this is sometime confusing. Also, they should be consistent with abbreviations. For example, cytochrome P450 is sometimes abbreviated to CYP, sometimes to CYP450. They should choose only one abbreviation per name.
  • Figure 1B: the font used in the figure does not allow a clear reading of the text. Maybe authors should use Helvetica or Arial.
  • Caption Figure 1:
  • write the full name of FADS2 and DPA;
  • “PUFA biosynthesistic pathway”; “1B)”.
  • Figure 2: Authors should indicate in the figure the 7 steps that are described in the text.
  • Caption Figure 4:
  • “alpha/beta fold”, please use greek letters;
  • add some spaces: “The she dimer isantiparallel…”; “…theCterminal region of the other”; “…the epoxide hydrolase activity islocated within….”;
  • “…., calculated from a gene model with isoforms…”.

Author Response

We want to thank the referees for their constructive comments and suggestions on the manuscript entitled “Cytochrome P450 metabolism of polyunsaturated fatty acids and neurodegeneration” (Manuscript ID: nutrients-969179). We have made careful revisions based on the feedback, and detailed response to the comments is attached.
